# Must All Negatives Be Pushed Away Equally? Uncertainty-Aware Cross-View Geo-Localization via Normal Inverse Gamma Distribution

**Songsong Ouyang** [1]  **Le Wu** [1]  **Yingying Zhu** [1]

## Abstract

Cross-view geo-localization (CVGL) aims to retrieve the corresponding satellite image given a street query and is critical for autonomous navigation. Although recent methods perform well on benchmarks, they often fail to generalize to unseen environments. A key limitation is the use of contrastive learning, which assigns equal labels to all negative samples and induces similarity-amplified repulsion. But should all negatives be treated equally? In CVGL, semi-positive samples that are geographically proximate to the positive often share important semantic cues. Treating them as ordinary negatives forces the model to overfit noise, leading to a collapse in generalization. To address this issue, we propose an uncertainty-aware framework grounded in Deep Evidential Regression (DER), modeling the Normal-Inverse-Gamma (NIG) distribution as a conjugate prior to quantify environmental complexity $u$ in a single forward pass. The estimated $u$ adaptively softens labels for hard negatives in Soft InfoNCE, mitigating excessive repulsion on semi-positive samples. An Uncertainty Head with cls-to-spatial cross-attention and attention statistics is designed to accurately fit the NIG distribution. Extensive experiments demonstrate state-of-the-art performance, including an average 18% R@1 improvement in zero-shot cross-dataset transfer, filling the critical gap between laboratory benchmarks and robust real-world deployment.

## 1. Introduction

Cross-view geo-localization aims to determine the geographic location by matching a street image with its cor-

[1]Shenzhen University, College of Computer Science and Software Engineering, Shenzhen, China. Correspondence to: Yingying Zhu <zhuyy@szu.edu.cn>.

*Proceedings of the 43rd International Conference on Machine Learning*, Seoul, South Korea. PMLR 306, 2026. Copyright 2026 by the author(s).

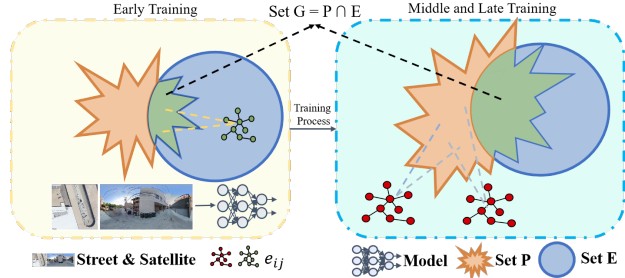

*Figure 1.* Set-theoretic Evolution of Generalization in Cross-View Geo-Localization. In the early stage, the model rapidly learns informative $e_{ij}$ that capture discriminative localization cues, leading to fast growth of localization capability. In later stages, equal negative labels over-separate hard negatives, causing the model to fit noisy parts of $e_{ij}$ and expand $P$ toward irrelevant regions, while the intersection $G = P \cap E$ stagnates.

responding satellite image. This task plays a critical role in many domains (Xu et al., 2025; Yao et al., 2024). While most existing methods (Ouyang & Zhu, 2025; Zhang & Zhu, 2024a) achieve strong performance on benchmark datasets, they often suffer from a hidden Generalization Collapse when deployed in unseen urban environments. We argue that the root cause lies in the conventional contrastive learning paradigm. When semi-positive samples exist, these methods tend to over-separate visually similar negatives, causing the model to overfit noise in the training set and substantially degrading its generalization performance.

To better understand the overfitting behavior of models in CVGL, we formulate a set-based framework for the CVGL task, as illustrated in Fig. 1. Specifically, we abstract the localization capability learned by a model on the training set as a set $P$, while the capability required to correctly localize samples in the evaluation set is denoted as set $E$. The generalization of the model is then defined as the intersection $G = P \cap E$. We further denote the information contained in a street–satellite image pair as $e_{ij}$ ($1 \le i, j \le B$), which includes both localization-relevant information and noise. During the early stage of training, street and satellite image pairs are fed into the model to expand $P$, with the expectation of simultaneously increasing the intersection $G$. At this stage, since the model is still immature, the learned informa-

tion is dominated by localization cues, and $G$ grows accordingly. However, when semi-positive samples are present, contrastive learning assigns them disproportionately large gradients in the mid-to-late stages of training, forcing the model to push away samples that overlap with the positive region. This behavior drives the model to fit noise around the positives, causing $P$ to expand toward unreliable regions while the intersection $G$ stagnates.

Most existing methods do not explicitly account for this issue, and instead rely on data augmentation (Zhang et al., 2024; Deuser et al., 2023; Wu et al., 2026; Zhang et al., 2026a;b; Li & Zhu, 2025) or causal learning mechanisms (Ouyang & Zhu, 2025) to force the model to continue learning or to reduce spurious correlations, thereby expanding the intersection $G$ to improve generalization. However, such strategies provide only limited benefits on datasets containing semi-positive samples. The core issue stems from assigning equal labels to all negative samples in contrastive learning, which causes hard negatives that share substantial semantic and geographic overlap with positives to be overly pushed away, leading the model to fit noise and ultimately harming generalization (Guo et al., 2024; Wang et al., 2024). To steer the expansion of $P$ back to the informative regions of $G$, we propose an uncertainty-aware framework grounded in Deep Evidential Regression (DER). Specifically, we model the Normal-Inverse-Gamma (NIG) distribution as a conjugate prior over the parameters of a Gaussian likelihood, enabling us to quantify the environmental complexity $u$ surrounding each positive sample in a single forward pass. To alleviate the excessive push-away effect caused by large gradients from semi-positive samples, we further design a Soft InfoNCE loss to mitigate noise overfitting, and employ an Uncertainty Head based on cls-to-spatial cross-attention and four attention-weight statistics to accurately fit the NIG distribution of positive samples.

In summary, our contributions are summarized as :

- We are the first to introduce Deep Evidential Regression to the CVGL. By fitting a higher-order NIG prior, the model can compute the neighborhood complexity of positive samples with a single forward pass. This approach is lightweight, simple, and effective.

- We design an Uncertainty Head that leverages cls-to-spatial cross-attention and four statistical measures of the attention weights to more accurately characterize the internal information of features, enabling the fitting of a correct NIG distribution.

- We are the first to analyze how contrastive learning with equal negative labels degrades generalization in CVGL under semi-positive samples. To address this, we introduce a Soft InfoNCE loss that alleviates the over-separation of semi-positive pairs, which is

strongly validated by large gains in zero-shot cross-dataset transfer.

Our analysis highlights that, in CVGL with semi-positive samples, uniformly pushing away hard negatives is ineffective and theoretically biased. By combining Deep Evidential Regression with Soft InfoNCE, we offer a simple yet effective solution that is readily extensible to other contrastive learning tasks facing similar semi-positive ambiguity.

## 2. Related Work

### 2.1. Cross-view Geo-localization

**Encoder.** In cross-view geo-localization (CVGL), (Workman et al., 2015) first showed that CNN-based features significantly outperform hand-crafted representations, ushering CVGL into the deep learning era (Hu et al., 2018; Wang et al., 2023). Following the introduction of Transformers (Vaswani et al., 2017), Transformer-based encoders gradually replaced CNNs. L2LTR (Yang et al., 2021) first applied Transformers to CVGL. TransGeo (Zhu et al., 2022) advanced this by using a pure Transformer architecture and attention-guided non-uniform cropping on satellite images, reducing computation and discarding irrelevant regions. Sample4Geo (Deuser et al., 2023) later introduced ConvNeXt (Liu et al., 2022) to CVGL, achieving strong performance. In this work, we adopt DINOv3 ViT-B (Siméoni et al., 2025) as our encoder due to its dense feature discriminability and global attention mechanism, which facilitate accurate modeling of the NIG distribution.

**Loss Function.** Since CVGL is fundamentally an image retrieval task aimed at matching street with satellite image, the majority of existing methods rely on triplet loss or its soft variants to optimize feature discriminability (Yang et al., 2021; Zhang & Zhu, 2024a; Zhu et al., 2022; Zhang et al., 2023). Sample4Geo (Deuser et al., 2023) instead introduced the InfoNCE loss (van den Oord et al., 2018) to further enhance the ability to pull positive pairs closer while pushing negative pairs apart, resulting in substantial performance improvements. To address the excessively large gradients induced by semi-positive samples, which may cause the model to over-separate these samples and overfit noise, we design a Soft InfoNCE loss to alleviate this issue.

### 2.2. Deep Evidential Regression

Although modern deep learning models have achieved remarkable performance across various downstream tasks, most of them make predictions without explicitly modeling confidence. Data noise and model overconfidence can severely degrade generalization performance, making uncertainty estimation particularly important for CVGL tasks. Most existing approaches estimated uncertainty via

Bayesian neural networks (Franchi et al., 2024), which relied on Monte Carlo sampling and multiple forward passes to predict the mean and variance under out-of-distribution (OOD) scenarios (Chun et al., 2021; Lu et al., 2023), resulting in substantial computational overhead. In this work, we adopt a deep evidential regression (Amini et al., 2020; Lou et al., 2023) framework with higher-order priors to obtain uncertainty estimates through a single forward inference, significantly reducing computational complexity. Recent works (Wang et al., 2025) introduce uncertainty estimation to penalize overconfident predictions. In contrast, to address noise-induced overfitting in CVGL, we leverage uncertainty $u$ to suppress excessively large gradients from semi-positive samples, substantially improving zero-shot cross-dataset generalization.

## 3. Method

To constrain the growth direction of $P$, it is insufficient to rely on hard-coded contrastive supervision, as equal negative labels fail to reflect the true structure of real-world retrieval scenarios and encourage the model to fit noisy $e_{ij}$ by over-separating semi-positive samples. To address this, we model the similarity of positive samples with a Gaussian likelihood equipped with a Normal Inverse Gamma (NIG) prior, which serves as a higher-order uncertainty-aware representation beyond point estimates. The uncertainty $u$ derived from the predicted NIG distribution naturally captures the ambiguity of a positive sample's local neighborhood, reflecting the environmental complexity induced by semi-positive samples. Based on this uncertainty, we design a Soft InfoNCE loss that adaptively moderates the repulsion strength on hard negatives, preventing noise overfitting while preserving effective discrimination.

### 3.1. Uncertainty Estimation

In Bayesian inference, estimating these parameters requires integrating over the posterior $p(\mu_i, \sigma_i^2 \mid \{t_{ij}\})$, which is analytically intractable for arbitrary priors. However, by adopting a Normal-Inverse-Gamma (NIG) conjugate prior, the posterior remains in the NIG family, enabling closed-form computation. The NIG prior is parameterized as:

$$\mu_i \mid \sigma_i^2 \sim \mathcal{N}(\gamma_i, \sigma_i^2/\nu_i), \quad \sigma_i^2 \sim \text{Inv-Gamma}(\alpha_i, \beta_i), \tag{1}$$

where $\{\gamma_i, \nu_i, \alpha_i, \beta_i\}$ are predicted by a neural network from the visual features of sample $i$. Under this formulation, the marginal distribution $p(t_{ij} \mid \gamma_i, \nu_i, \alpha_i, \beta_i)$ follows a Student-$t$ distribution.

**Normal Inverse Gamma Distribution.** Specifically, given an observation $y$, we assume that it is drawn from a Gaussian distribution $y \sim \mathcal{N}(\mu, \sigma^2)$, and impose an NIG prior over the parameters $(\mu, \sigma^2)$:

$$\begin{aligned} \mu &\sim \mathcal{N}(\gamma, \sigma^2/\nu), \\ \sigma^2 &\sim \Gamma^{-1}(\alpha, \beta), \end{aligned} \tag{2}$$

where $\Gamma^{-1}(\cdot)$ denotes the inverse-Gamma distribution, $\{\gamma, \nu, \alpha, \beta\}$ are the four parameters of the NIG distribution, which characterize predictive uncertainty. By the conjugacy property, the posterior distribution can be written as:

$$\begin{aligned} p(\mu, \sigma^2 \mid \gamma, \nu, \alpha, \beta) = &\frac{\beta^\alpha \sqrt{\nu}}{\Gamma(\alpha)\sqrt{2\pi\sigma^2}} \left(\frac{1}{\sigma^2}\right)^{\alpha+1} \\ &\times \exp\left(-\frac{1}{2\sigma^2}\left(2\beta + \nu(\gamma-\mu)^2\right)\right). \end{aligned} \tag{3}$$

Specifically, the NIG parameters admit an intuitive interpretation via virtual observations: $\gamma$ represents the predicted mean, $\nu$ encodes the precision of this estimate (higher $\nu$ indicates stronger confidence), $\alpha$ quantifies the amount of evidence supporting the variance estimate, and $\beta$ controls the scale of variance. Under this interpretation, the NIG prior jointly encodes evidence about both the mean and the variance. Consequently, the total amount of evidence accumulated by the model is quantified by $2\nu + \alpha$, which aggregates the virtual observations supporting the estimation of $\mu$ and $\sigma^2$. A larger total evidence corresponds to higher confidence and lower predictive uncertainty.

The expectations of the NIG distribution admit closed-form expressions:

$$\mathbb{E}[\mu] = \gamma, \quad \mathbb{E}[\sigma^2] = \frac{\beta}{\alpha-1}, \quad \text{Var}[\mu] = \frac{\beta}{\nu(\alpha-1)}. \tag{4}$$

Here, $\mathbb{E}[\sigma^2]$ represents the aleatoric uncertainty, capturing the inherent noise of the observations, while $\text{Var}[\mu]$ corresponds to the epistemic uncertainty, reflecting uncertainty in the model's prediction due to limited evidence. In our method, aleatoric uncertainty is the dominant component, with epistemic uncertainty serving as a complementary term that diminishes as the model converges. This total uncertainty also reflects the complexity of the environment surrounding the sample. Critically, $u_i$ is computed in a single forward pass, avoiding the computational overhead of Monte Carlo sampling. We define the total predictive uncertainty as the sum of these two components:

$$u = \mathbb{E}[\sigma^2] + \text{Var}[\mu] = \frac{\beta(\nu+1)}{\nu(\alpha-1)}. \tag{5}$$

### 3.2. NIG for Cross-view Geo-localization

In CVGL, the presence of semi-positive samples means that forcibly pushing such samples apart may cause the model to overfit to noise. Therefore, it is necessary to introduce uncertainty to assess the environmental complexity of the reference images involved in the current pairing.

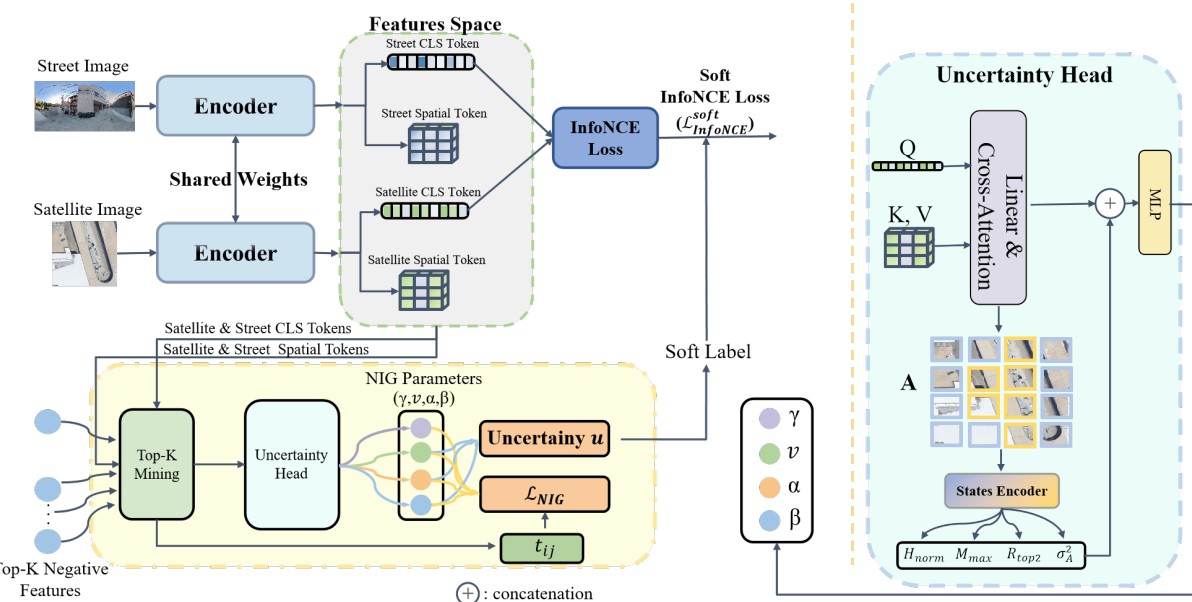

*Figure 2.* Overall framework of the proposed model. For each positive sample, candidate negative reference images are first ranked by similarity to the query image, $s_{ij}$, and the Top-$K$ negatives are selected. The positive sample's similarities to these Top-$K$ negatives, $t_{ij}$, serve as observations to train the Uncertainty Head, which predicts the four parameters of the NIG distribution to model the positive sample's neighborhood complexity. The resulting uncertainty $u_i$ is then used to generate soft labels, mitigating the excessive repulsion of hard negatives during training.

Then, we quantify the uncertainty of each positive sample by modeling the distribution of its neighborhood similarity. Specifically, for a positive sample $i$, we observe $K$ same-modal similarities $\{t_{ij}\}_{j \in \mathcal{N}_K(i)}$ to its Top-$K$ hard negatives $\mathcal{N}_K(i)$, where $t_{ij} = \text{cosine}(\mathbf{v}_i, \mathbf{v}_j)$ measures the similarity between candidate reference images. Following Deep Evidential Regression, we assume these observations are drawn from an approximately Gaussian distribution:

$$t_{ij} \mid \mu_i, \sigma_i^2 \sim \mathcal{N}(\mu_i, \sigma_i^2), \quad \text{cosine}(\mathbf{v}_i, \mathbf{v}_j) = \frac{\mathbf{v}_i \cdot \mathbf{v}_j}{\|\mathbf{v}_i\| \|\mathbf{v}_j\|}. \tag{6}$$

where $\mathbf{v}_i$ and $\mathbf{v}_j$ denote the feature representations derived from the cls tokens of the image encoder for the positive and hard negative reference images, respectively. $\|\cdot\|$ means the L2 normalization. $\mu_i$ and $\sigma_i^2$ are unknown parameters specific to sample $i$. Therefore, we adopt a NIG distribution as the prior of $\mathcal{N}(\mu_i, \sigma_i^2)$, and compute the uncertainty $u$ from the predicted NIG parameters. The estimated uncertainty is further used to generate soft labels and implement a soft InfoNCE objective, thereby constraining the growth of $P$ toward informative regions and improving the generalization ability $G = P \cap E$.

**Uncertainty Head.** To characterize an NIG distribution that can reflect the environmental complexity in which a positive sample resides, the design of the uncertainty head is critical. We use positive sample's cls token as the query and the spatial tokens as keys and values, both extracted from

the same view (either street or satellite), to approximate the neighborhood complexity, enabling the model to focus on the spatial distribution of high-level semantic information. Based on the attention weights $\mathbf{A} = \text{Softmax}(\mathbf{Q}\mathbf{K}^\top / \sqrt{d_h})$, we extract four statistical measures: normalized entropy $H_{\text{norm}}$ (dispersion), maximum response $M_{\text{max}}$ (salience), Top-2 ratio $R_{\text{top2}}$ (multi-modality), and variance $\sigma_A^2$ (non-uniformity). These statistics capture the global spatial layout of positive images. The attention outputs and the extracted statistics are concatenated and fed into an MLP to predict the NIG hyperparameters. Using the similarity $t_{ij}$ as observations, the Uncertainty Head is trained by the NIG loss to learn an accurate NIG distribution and thus estimate the $u$. The head architecture is illustrated on the right side of Fig. 2.

$$H_{\text{norm}} = \frac{-\sum_i A_i \ln(A_i + \epsilon)}{\ln(HW)}, \quad M_{\text{max}} = \max(\mathbf{A}),$$

$$R_{\text{top2}} = \frac{\text{top}_2(\mathbf{A})}{\max(\mathbf{A}) + \epsilon}, \quad \sigma_A^2 = \text{Var}(\mathbf{A}), \tag{7}$$

$$\mathbf{s} = [H_{\text{norm}}, M_{\text{max}}, R_{\text{top2}}, \sigma_A^2],$$

$$\{\hat{\gamma}, \hat{\nu}, \hat{\alpha}, \hat{\beta}\} = \text{MLP}(\text{concat}(\mathbf{z}, \text{Linear}(\mathbf{s}))).$$

Here, $\mathbf{z}$ denotes the attention output. $H$ and $W$ denote the spatial dimensions of the feature map, and $\text{top}_2(\mathbf{A})$ returns the second largest value in the attention vector $\mathbf{A}$. $\epsilon > 0$ is a tiny constant. $\{\hat{\gamma}, \hat{\nu}, \hat{\alpha}, \hat{\beta}\}$ define the NIG distribution

$\text{NIG}(\mu, \sigma^2 \mid \gamma, \nu, \alpha, \beta)$, which serves as a prior over the Gaussian likelihood parameters $(\mu, \sigma^2)$. We apply a softplus activation to ensure all parameters are positive, and additionally add an offset of 1 to $\alpha$ to enforce $\alpha > 1$. The uncertainty is computed as shown in Eq. 5.

**Soft Label.** In the presence of semi-positive samples, standard InfoNCE with label smoothing (0.1) assigns equal labels to all negatives, causing hard negatives that are more similar to the query to receive disproportionately large gradients and be forcibly pushed away. This leads the model to overfit noise and reduces generalization, motivating the introduction of soft labels. To ensure that the estimated uncertainty $u_i$ accurately reflects the environmental complexity around a positive sample, we first select the Top-$K$ negative candidates based on the similarity $s_{ij}$ between the query and negative reference samples, ensuring that these negatives are geographically adjacent rather than visually similar due to shared styles. Ordinary negatives outside the Top-$K$ are assigned a base label $\lambda_{\text{base}} = 0.1/(B-1)$, where $B$ denotes the batch size, preserving consistency with label smoothing. The $u_i$ are then used to scale the negative labels in InfoNCE, reducing the repulsion on hard negatives while maintaining separation for ordinary negatives.

Based on the uncertainty $u_i$ and the similarity $s_{ij}$ between the Top-$K$ reference negative samples and the query feature, with $j \in \mathcal{N}_K(i)$, $\lambda_{base}$ is the baseline weight, the soft label is defined as:

$$\lambda_j^{soft} = \lambda_{\text{base}} + s_{ij} \cdot u_i \cdot w, \qquad (8)$$

where $w$ controls smoothing strength. Through $w$, we ensure that easy negatives within the Top-$K$ are pushed away with the same strength as ordinary negatives, while hard negatives are neither overly pulled closer nor aggressively pushed away, but instead experience a moderated push-away force. The positive sample label is then defined as one minus the sum of all negative sample labels. Given that $u < 1$ and $w$ is set to a sufficiently small value, the resulting positive label is guaranteed to be strictly greater than 0.

### 3.3. Loss Function

Given the similarity $t_{ij}$ and the NIG parameters, the negative log-likelihood (NLL) loss is defined as:

$$
\begin{aligned}
\mathcal{L}_{NLL}^i = & -\log p(t_{ij} \mid \gamma_i, \nu_i, \alpha_i, \beta_i) \\
= & \frac{1}{2} \log\left(\frac{\pi}{\nu_i}\right) - \alpha_i \log \Omega_i \\
& + \left(\alpha_i + \frac{1}{2}\right) \log\left((t_{ij} - \gamma_i)^2 \nu_i + \Omega_i\right) \\
& + \log \Gamma(\alpha_i) - \log \Gamma\left(\alpha_i + \frac{1}{2}\right),
\end{aligned}
\qquad (9)
$$

where $\Omega_i = 2\beta_i(1 + \nu_i)$. This loss maximizes the model evidence with respect to the observed data. To prevent the model from becoming overconfident in misleading evidence, we introduce a regularization term to penalize erroneous evidence:

$$\mathcal{L}_{REG}^i = (t_{ij} - \gamma_i)^2 \cdot \left(2\nu_i + \alpha_i + \frac{1}{\beta_i}\right), \qquad (10)$$

The $1/\beta$ term provides adaptive modulation—when $\beta$ is large (high uncertainty), $1/\beta$ becomes small, reducing the penalty and allowing the model to maintain high uncertainty. The overall NIG loss is given by:

$$\mathcal{L}_{\text{NIG}} = \frac{1}{B \cdot K} \sum_{i=1}^{B} \sum_{j=1}^{K} \left(\mathcal{L}_{\text{NLL}}^i + \lambda_{\text{reg}} \mathcal{L}_{\text{REG}}^i\right), \qquad (11)$$

where $K$ denotes the number of Top-$K$ samples.

**Uncertainty-Aware Soft InfoNCE.** Standard InfoNCE assigns uniform gradients to all negatives via hard one-hot labels. For hard negatives with high similarity $s_{ij} \approx s_{ii}$, the gradient $\partial \mathcal{L} / \partial s_{ij} \propto \exp(s_{ij}/\tau)$ becomes excessively large, aggressively pushing semantically similar samples apart. Even with label smoothing, semi-positive samples are pushed away more aggressively due to their higher similarity to the query. In CVGL, some datasets naturally include semi-positive samples, where a single query has one positive match and additional hard negatives that are geographically proximate and semantically similar. In such cases, forcibly pushing away all negative samples can cause the model to overfit to noise, leading the training capacity set $P$ to expand toward unreliable regions. As a result, the intersection $G = P \cap E$ reaches a performance bottleneck, and the fine-grained geographic discriminability of the model is consequently degraded.

To address this, following the definition in Eq. 8, we assign labels to the Top-$K$ negative samples for each positive.:

$$y_{ij}^{soft} = \begin{cases} 1 - \sum_{k \neq i} y_{ik}^{soft}, & j = i, \\ \lambda_j^{soft}, & j \in \mathcal{N}_K(i), \\ \lambda_{base}, & \text{otherwise}, \end{cases} \qquad (12)$$

Let $p_{ij} = \frac{\exp(s_{ij}/\tau)}{\sum_{k=1}^{B} \exp(s_{ik}/\tau)}$ be the predicted probability. The soft InfoNCE loss:

$$\mathcal{L}_{InfoNCE}^{soft} = -\frac{1}{B} \sum_{i=1}^{B} \sum_{j=1}^{B} y_{ij}^{soft} \cdot \log p_{ij} \qquad (13)$$

By assigning $y_{ij}^{\text{soft}} = \lambda_{\text{base}} + s_{ij} \cdot u_i \cdot w$ to hard negatives, the gradient becomes:

$$\frac{\partial \mathcal{L}^{\text{soft}}}{\partial s_{ij}} = p_{ij} - y_{ij}^{\text{soft}}, \qquad (14)$$

*Table 1.* Comparisons with state-of-the-art models on the CVUSA, CVACT_val and CVACT_test datasets. († denotes models that use the polar transformation.)

| MODEL | CVUSA | | | | CVACT_VAL | | | | CVACT_TEST | | | |
|---|---|---|---|---|---|---|---|---|---|---|---|---|
| | R@1 | R@5 | R@10 | R@1% | R@1 | R@5 | R@10 | R@1% | R@1 | R@5 | R@10 | R@1% |
| LPN (WANG ET AL., 2022) | 85.79 | 95.38 | 96.98 | 99.41 | 79.99 | 90.63 | 92.56 | - | - | - | - | - |
| LPN† (WANG ET AL., 2022) | 92.83 | 98.00 | 98.85 | 99.78 | 83.66 | 94.14 | 95.92 | 98.41 | - | - | - | |
| DSM (SHI ET AL., 2020) | 91.96 | 97.50 | 98.54 | 99.67 | 82.49 | 92.44 | 93.99 | 97.32 | - | - | - | |
| TRANSGEO (ZHU ET AL., 2022) | 94.08 | 98.36 | 99.04 | 99.77 | 84.95 | 94.14 | 95.78 | 98.37 | - | - | - | |
| GEODTR (ZHANG ET AL., 2023) | 93.76 | 98.47 | 99.22 | 99.85 | 85.43 | 94.81 | 96.11 | 98.26 | 62.96 | 87.35 | 90.70 | 98.61 |
| GEODTR+ (ZHANG ET AL., 2024) | 95.05 | 98.42 | 98.92 | 99.77 | 87.76 | 95.50 | 96.50 | 98.32 | 67.75 | 90.15 | 92.73 | 98.53 |
| GEODTR† (ZHANG ET AL., 2023) | 95.43 | 98.86 | 99.34 | 99.86 | 86.21 | 95.44 | 96.72 | 98.77 | 64.52 | 88.59 | 91.96 | 98.74 |
| SAMPLE4GEO (DEUSER ET AL., 2023) | 98.68 | 99.68 | 99.78 | 99.87 | 90.81 | 96.74 | 97.48 | 98.77 | 71.51 | 92.42 | 94.45 | 98.70 |
| CONGEO (MI ET AL., 2024) | 98.30 | - | - | 99.90 | 90.10 | - | - | 98.20 | 71.70 | 98.30 | - | - |
| EP-BEV (YE ET AL., 2024) | 97.41 | 99.40 | 99.60 | 99.76 | 90.61 | 96.57 | 97.32 | 98.71 | 71.41 | 92.38 | 94.37 | 98.77 |
| CLGT (OUYANG & ZHU, 2025) | 98.85 | 99.71 | 99.81 | 99.86 | 91.97 | 96.95 | 97.72 | 98.77 | 73.22 | 93.50 | 95.23 | 98.79 |
| OURS (UA-GEO) | **99.30** | **99.82** | **99.84** | **99.93** | **92.87** | **97.77** | **98.21** | **99.11** | **73.63** | **94.38** | **95.94** | **99.05** |

*Table 2.* Performance comparison on VIGOR.

| MODEL | SAME-AREA | | | | | CROSS-AREA | | | | |
|---|---|---|---|---|---|---|---|---|---|---|
| | R@1 | R@5 | R@10 | R@1% | HIT RATE | R@1 | R@5 | R@10 | R@1% | HIT RATE |
| TRANSGEO (ZHU ET AL., 2022) | 61.48 | 87.54 | 91.88 | 99.56 | 73.09 | 18.99 | 38.24 | 46.91 | 88.94 | 21.21 |
| SAIG-D (ZHU ET AL., 2023) | 65.23 | 88.08 | - | 99.68 | 74.11 | 33.05 | 55.94 | - | 94.64 | 36.71 |
| GEODTR+ (ZHANG ET AL., 2024) | 59.01 | 81.77 | 87.10 | 99.07 | 67.41 | 36.01 | 59.06 | 67.22 | 94.95 | 39.40 |
| EP-BEV(384 × 384) (YE ET AL., 2024) | 76.08 | 94.71 | 96.59 | 99.64 | 87.01 | 60.03 | 81.63 | 86.34 | 97.64 | 67.88 |
| SAMP4GEO (DEUSER ET AL., 2023) | 77.86 | 95.66 | 97.21 | 99.61 | 89.82 | 61.70 | 83.50 | 88.00 | 98.17 | 69.87 |
| CLGT (OUYANG & ZHU, 2025) | 77.18 | 94.57 | 96.42 | 99.54 | 87.95 | 60.70 | 82.36 | 86.92 | 97.82 | 68.40 |
| OURS | **79.48** | **96.41** | **97.62** | **99.72** | **91.60** | **66.70** | **87.79** | **91.37** | **98.69** | **76.46** |

where $p_{ij} = \exp(s_{ij}/\tau) / \sum_{k=1}^{B} \exp(s_{ik}/\tau)$. The scale $w$ modulates the soft label magnitude based on both similarity and uncertainty: by setting $y_{ij}^{\text{soft}} \propto s_{ij} \cdot u_i$, hard negatives with high $s_{ij}$ and $u_i$ receive larger target probabilities, thereby reducing the gradient $|p_{ij} - y_{ij}^{\text{soft}}|$ and mitigating excessive repulsion, while easy negatives with low $s_{ij}$ retain minimal labels $\approx \lambda_{\text{base}}$, preserving discrimination. This strategy reduces noise fitting, constraining the growth of $P$ toward reliable regions and thereby maximizing $G$. The final training objective is:

$$\mathcal{L}_{\text{total}} = \mathcal{L}_{InfoNCE}^{soft} + \lambda_{\text{nig}} \cdot \mathcal{L}_{\text{NIG}}. \quad (15)$$

# 4. Experiment

**Dataset.** We evaluate our model on three widely-used cross-view geo-localization benchmarks—CVUSA (Workman et al., 2015), CVACT (Liu & Li, 2019), and VIGOR (Zhu et al., 2021). CVUSA and CVACT each provide 35,532 training and 8,884 testing image pairs with a strict 1-to-1 ground-to-aerial correspondence. CVACT offers an extra 92,802 GPS-tagged query images for large-scale retrieval evaluation, making it suitable for both standard and large-scale testing scenarios. VIGOR is a more challenging benchmark that spans four metropolitan areas—New York, Seattle, San Francisco, and Chicago—and includes 105,214 query and 90,618 reference images. Unlike CVUSA and CVACT, VIGOR introduces a harder retrieval setup by assigning each query one true positive and three semi-positive samples, thus increasing the difficulty of discriminative matching.

**Evaluation Protocol.** We adopt Recall@K as the primary evaluation metric, where $K \in \{1, 5, 10\}$, as well as Recall@1%. A query is considered correctly localized if its corresponding aerial image appears among the top-K retrieved candidates for a given street panorama.

## 4.1. Implementation Details

To better suit our framework and the CVGL task, we adopt DINOv3-ViT-Base as the encoder during training to obtain global attention maps. Our baseline follows Sample4Geo framework with a DINOv3-ViT-B encoder. An ablation study on different encoders is reported in Table.3. The model is optimized using AdamW (Loshchilov & Hutter, 2017) with an initial learning rate of $1 \times 10^{-4}$ and trained for 40 epochs with a batch size of 128. The regularization weight $\lambda_{reg}$ and the weight $\lambda_{nig}$ for the NIG loss are set to 0.01 and 0.5, respectively. Hyperparameter studies on Top-$K$ and the weighting factor $w$ are presented in Table.4. During training, we first warm up the encoder for 4 epochs with the Uncertainty Head frozen, ensuring stable feature representations. After the encoder stabilizes, the Uncertainty Head is jointly trained with the encoder for 3 warm-up epochs to learn reliable NIG predictions.

## 4.2. Comparing with State-of-the-art Models

**Cross-view Image retrieval.** To evaluate the effectiveness of our method, we conduct experiments on several benchmark datasets. Table. 1 reports our results on CVACT and

*Table 3.* Ablation studies on the VIGOR Same dataset. We evaluate the impact of the Uncertainty Head (top) and different Encoders (bottom).

| Variant | R@1 | R@5 | R@10 | R@1% |
|---|---|---|---|---|
| *Impact of Uncertainty Head* | | | | |
| Ours (default) | **79.48** | **96.41** | **97.62** | **99.72** |
| w/o States | 78.92 | 96.22 | 97.34 | 99.67 |
| *Impact of Encoder (Sample4Geo Framework)* | | | | |
| ConvNeXt-B | **77.86** | 95.66 | **97.21** | 99.61 |
| DINOv3-ViT-B | 77.68 | **95.98** | 97.18 | **99.65** |

*Table 4.* Hyperparameter studies on the VIGOR Same dataset. We analyze the impact of Top-K candidates (top) and the weight parameter $w$ (bottom).

| Value | R@1 | R@5 | R@10 | R@1% |
|---|---|---|---|---|
| *Impact of Top-K ($K$)* | | | | |
| 3 | 78.68 | 96.38 | 97.51 | 99.64 |
| 5 | **79.48** | **96.41** | 97.62 | **99.72** |
| 8 | 79.09 | 96.24 | **97.66** | 99.69 |
| *Impact of Weight ($w$)* | | | | |
| 0.05 | 78.38 | 96.15 | 97.41 | 99.63 |
| 0.1 | **79.48** | **96.41** | **97.62** | **99.72** |
| 0.3 | 78.69 | 96.33 | 97.42 | 99.66 |

CVUSA, where we achieve state-of-the-art performance. This prevents the model from overfitting noise by excessively separating inherently similar negative samples, resulting in strong performance on both CVACT_val and CVACT_test. For VIGOR, the similarity gap between semi-positive and positive pairs is narrower than that in CVACT and CVUSA, which limits the ability of the scaling factor $w$ to selectively relax semi-positive samples without affecting ordinary negatives. Despite this constraint, our method still achieves consistent gains, improving performance by 1.7% on VIGOR-Same and 5% on VIGOR-Cross, while maintaining comparable parameter count and computational complexity to the baseline. A detailed complexity analysis is provided in the Appendix.

**Transfer Learning.** To validate the core claim of this work—solving the Generalization Collapse, we conduct extensive cross-dataset transfer experiments in Table. 5. Our framework yields a staggering average improvement of 18% in R@1 across all transfer tasks. For instance, when trained on CVUSA and tested on CVACT_test, our model achieves 54.99% R@1, outperforming the previous best CLGT by over 19%. In particular, we observe that both CVACT and CVUSA contain semi-positive samples. CVACT exhibits extremely similar semi-positives, where the top-ranked reference image can reach a similarity of up to 0.9 with the positive, causing severe interference during retrieval. In contrast, CVUSA contains partially duplicated samples; when such samples co-occur within the same batch, forcibly pushing them apart also induces noise fitting. By alleviating the over-penalization of these semi-positive samples, our Soft InfoNCE effectively improves generalization, which explains the substantial performance gains reported in Table. 5. Detailed analyses, validations, and additional experiments are provided in the Appendix.

**Ablation Study.** We conduct ablation studies on the design of the encoder and the head, as shown in Table. 3. It can be observed that, under the Sample4Geo framework where the encoder is replaced, DINOv3-ViT-B achieves performance comparable to that of ConvNeXt-B. However, in the CVGL task, to accurately capture the neighborhood complexity of positive samples for fitting a precise NIG distribution, we adopt DINOv3-ViT-B as our encoder to better accommodate

*Table 5.* Results on different cross-dataset transfer tasks. († denotes models that use the polar transformation.)

| Method | R@1 | R@5 | R@1% |
|---|---|---|---|
| **CVUSA → CVACT_val** | | | |
| L2LTR (Yang et al., 2021) | 47.55 | 70.58 | 91.39 |
| L2LTR† (Yang et al., 2021) | 52.58 | 75.81 | 93.51 |
| GeoDTR (Zhang et al., 2023) | 47.79 | 70.52 | 92.20 |
| GeoDTR† (Zhang et al., 2023) | 53.16 | 75.62 | 93.80 |
| Samp4G (Deuser et al., 2023) | 56.62 | 77.79 | 94.69 |
| EP-BEV (Ye et al., 2024) | 59.32 | 80.79 | 94.69 |
| CLGT (Ouyang & Zhu, 2025) | 64.82 | 84.38 | 96.16 |
| Ours | **82.01** | **93.80** | **98.08** |
| **CVUSA → CVACT_test** | | | |
| L2LTR (Yang et al., 2021) | - | - | - |
| L2LTR† (Yang et al., 2021) | - | - | - |
| GeoDTR (Zhang et al., 2023) | 11.24 | 18.69 | 72.09 |
| GeoDTR† (Zhang et al., 2023) | 22.09 | 32.22 | 85.53 |
| Samp4G (Deuser et al., 2023) | 27.78 | 52.08 | 94.88 |
| EP-BEV (Ye et al., 2024) | 32.68 | 58.62 | 95.21 |
| CLGT (Ouyang & Zhu, 2025) | 35.52 | 63.37 | 96.35 |
| Ours | **54.99** | **83.46** | **98.00** |
| **CVACT → CVUSA** | | | |
| EP-BEV (Ye et al., 2024) | - | - | - |
| CLGT (Ouyang & Zhu, 2025) | - | - | - |
| Samp4G (Deuser et al., 2023) | 44.95 | 64.36 | 90.95 |
| GeoDTR+ (Zhang et al., 2024) | 52.56 | 73.08 | 94.80 |
| Ours | **69.34** | **83.39** | **96.89** |

the head design. As shown in Table. 3, removing the four attention-weight statistics leads to a drop in performance, indicating that these statistics provide valuable information for accurately fitting the $\text{NIG}(\mu, \sigma^2 \mid \gamma, \nu, \alpha, \beta)$.

**Hyperparameter Analysis.** To examine the effect of key hyperparameters, we study the scaling factor $w$ in Eq. 8 and the Top-$K$ selection. The scaling factor $w$ controls the relative gradient strength for semi-positive samples: under label smoothing (0.1), ordinary negatives receive a label of $0.1/(B-1)$, and we scale this by $w$ for hard negatives. We set $w = 0.1$ to sufficiently mitigate excessive pushaway without weakening positive sample learning; detailed derivation of this choice are provided in the Sec 5. The Top-$K$ parameter is set based on the typical number of semi-positive samples per positive in VIGOR , and choosing $K = 5$ ensures coverage of the neighborhood complexity while avoiding excessive inclusion of easy negatives. Table. 4 reports the corresponding performance.

**Visualization Analysis.** To provide an intuitive understand-

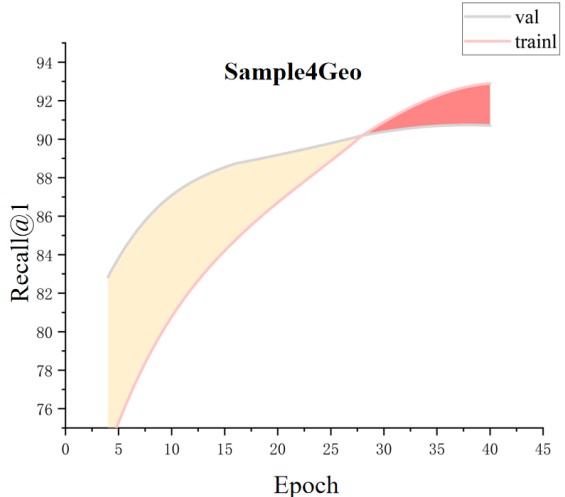

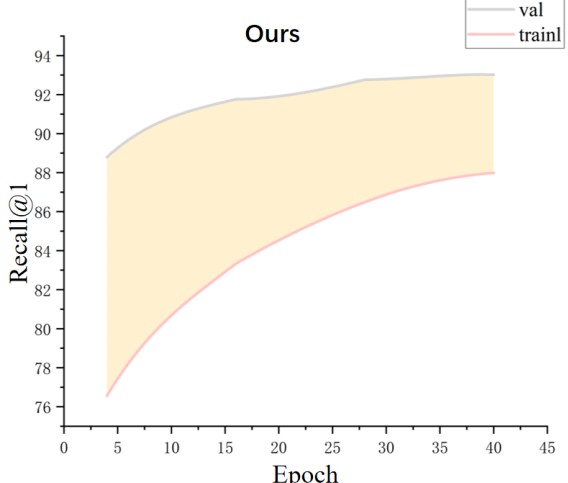

*Figure 3.* Training and validation Recall@1 of Sample4Geo on CVACT.

*Figure 4.* Training and validation Recall@1 of our method on CVACT.

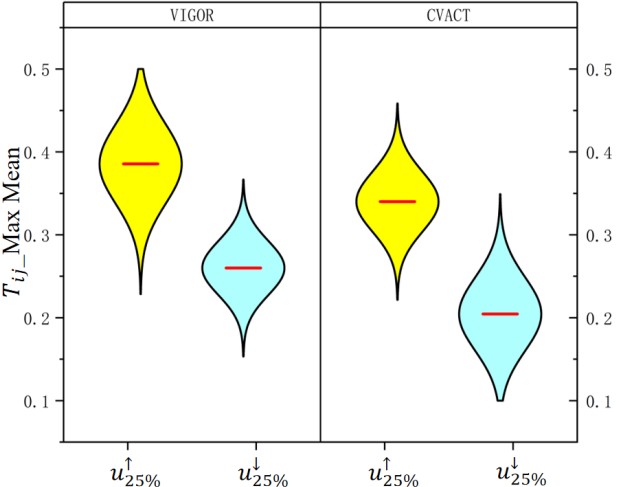

*Figure 5.* Violin plots of $u$ and the corresponding maximum $t_{ij}$ averages. For both CVACT and VIGOR, we record the average of the maximum $t_{ij}$ for the top 25% and bottom 25% of $u$ values.

ing of the effectiveness of our method, we visualize the relationship between uncertainty $u_i$ and the maximum same-modal similarity $t_{ij}$ during training on VIGOR and CVACT using violin plots in Fig. 5. We group samples by the top 25% and bottom 25% of $u_i$, and compute the average maximum $t_{ij}$ within each group. As shown, the top 25% $u$ samples exhibit approximately $1.5\times$ higher average maximum $t_{ij}$ compared to the bottom 25%. This validates that $u$ effectively captures local neighborhood complexity: when a positive sample has high same-modal similarity with nearby candidates, the NIG head produces larger $\beta$ to accommodate the higher variance in observed similarities, and smaller $\nu$ reflecting lower confidence in predicting a sin-

gle mean value, thereby increasing $u$ according to Eq. 5. Consequently, when these hard negatives also share high cross-modal similarity with the query, their soft labels are substantially increased via Eq. 8, mitigating excessive repulsion while preserving discrimination on easy negatives. Furthermore, as illustrated in Fig. 4 and Fig. 3, by mitigating the over-separation of semi-positive samples, our method reduces the likelihood of fitting training noise and markedly improves generalization, whereas forcibly pushing away such samples encourages noise fitting to reduce training loss, ultimately leading to overfitting.

**Gaussian Modeling of $t_{ij}$.** To validate the rationality of our assumption, we analyze the distribution of $t_{ij}$ on the VIGOR dataset, where $t_{ij}$ denotes the similarity between a positive sample $i$ and its Top-$K$ hard negatives. As shown in Fig. 6, both the density histogram and Quantile-Quantile (Q-Q) plot demonstrate that the similarity noise is well-approximated by a Gaussian distribution. This empirical observation supports our hypothesis that the conditional distribution for each individual positive sample follows a Gaussian form ($t_{ij} \sim \mathcal{N}(\mu_i, \sigma_i^2)$), thereby justifying the modeling of the Normal-Inverse-Gamma (NIG) prior. Furthermore, as the NIG loss primarily constrains the first and second moments, the framework remains robust to mild distributional shifts. Notably, $s_{ij}$ and $t_{ij}$ show no direct correlation.

## 5. Derivation of Soft Label Scale $w$

We derive the scale parameter $w$ in Eq. 8 to quantify the gradient mitigation for hard negatives. The gradient of the soft InfoNCE loss is:

$$\frac{\partial \mathcal{L}^{soft}}{\partial s_{ij}} = p_{ij} - y_{ij}^{soft}, \tag{16}$$

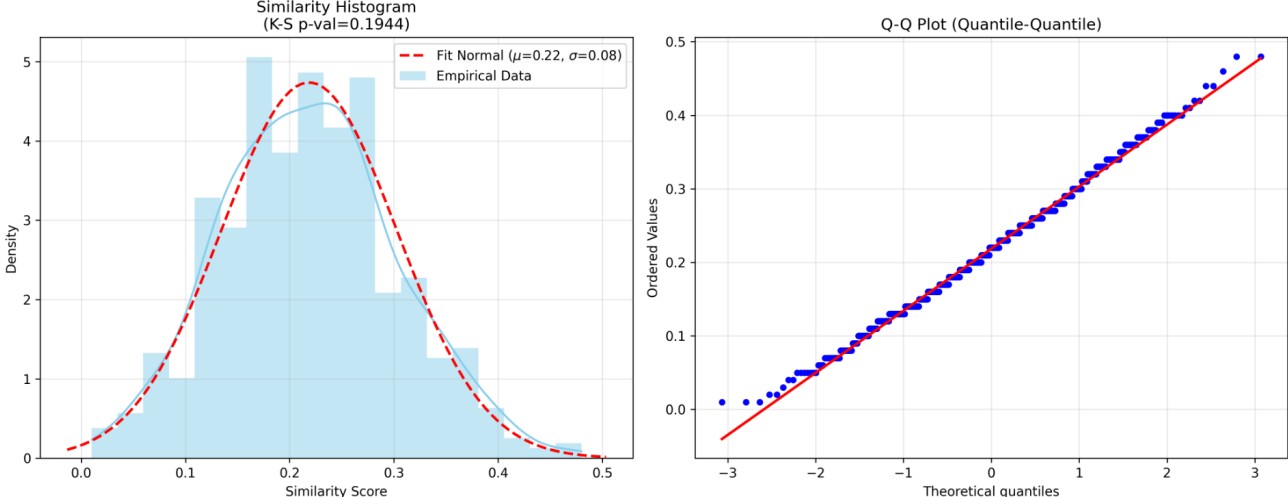

*Figure 6.* Statistical verification of the Gaussian assumption for $t_{ij}$. (Left) The density distribution of similarity scores $t_{ij}$, which approximates a Gaussian shape. (Right) The Quantile-Quantile (Q-Q) plot, where the close alignment of the data points (blue) with the theoretical diagonal line (red) statistically validates that $t_{ij}$ follows an approximately Gaussian distribution.

where $p_{ij} = \exp(s_{ij}/\tau)/\sum_{k=1}^{B} \exp(s_{ik}/\tau)$ and $y_{ij}^{soft} = \lambda_{base} + s_{ij} \cdot u_i \cdot w$ $(j \in \mathcal{N}_K(i))$. For hard negatives with maximum uncertainty $u_i \approx 0.5$ and similarity $s_{hard}$:

$$y_{hard}^{soft} = \lambda_{base} + s_{hard} \cdot 0.5 \cdot w. \qquad (17)$$

The gradient reduction from baseline to soft labels is:

$$\begin{aligned}
\Delta g &= (p_{hard} - \lambda_{base}) - (p_{hard} - y_{hard}^{soft}) \\
&= y_{hard}^{soft} - \lambda_{base} \\
&= s_{hard} \cdot 0.5 \cdot w.
\end{aligned} \qquad (18)$$

Define the gradient suppression ratio as:

$$\rho = \frac{p_{hard} - y_{hard}^{soft}}{p_{hard} - \lambda_{base}} = 1 - \frac{s_{hard} \cdot 0.5 \cdot w}{p_{hard} - \lambda_{base}}. \qquad (19)$$

Targeting $\rho \in [0.3, 0.5]$ (retaining 30%–50% of the original gradient), solving for $w$:

$$w = \frac{(1 - \rho)(p_{hard} - \lambda_{base})}{s_{hard} \cdot 0.5}. \qquad (20)$$

For CVGL with $\lambda_{base} \approx 0.001$, $p_{hard} \in [0.03, 0.08]$:

- **VIGOR** ($s_{hard} \approx 0.6$): Eq. 20 yields $w \in [0.05, 0.15]$. We set $w = 0.1$, achieving gradient suppression $\rho \approx 0.4$ via Eq. 19.

- **CVACT/CVUSA** ($s_{hard} \approx 0.9$): Eq. 20 yields $w \in [0.03, 0.10]$. We set $w = 0.05$, achieving $\rho \approx 0.45$.

This ensures hard negative gradients are reduced to 40%–50% of their baseline magnitude, mitigating excessive repulsion while preserving discrimination.

## 6. Conclusion and Future Work

In this work, we adopt deep evidential regression to fit an NIG distribution and estimate the environmental complexity of positive samples. This information is used to implement a Soft InfoNCE loss, mitigating the excessive repulsion of semi-positive samples and substantially improving model generalization. Experimental results demonstrate state-of-the-art performance, highlighted by a remarkable 18% average R@1 improvement in zero-shot cross-dataset transfer tasks. These findings validate that equal label assignment in contrastive learning leads to overfitting to noise, which negatively impacts generalization $G$. Future research will focus on adaptive mechanisms to further balance discriminative precision and robust uncertainty calibration.

## Acknowledgments

This work was supported in part Guangdong Basic and Applied Basic Research Foundation under Grant 2026A1515011137, in part by Shenzhen Science and Technology Program under Grant JCYJ20240813142510014 and Grant 202208810142553001, and in part by the Key Project of Department of Education of Guangdong Province under Grant 2023ZDZX1016.

## Impact Statement

This paper presents work whose goal is to advance the field of Machine Learning. There are many potential societal consequences of our work, none which we feel must be specifically highlighted here.

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

**Street image**                    **Satellite image**

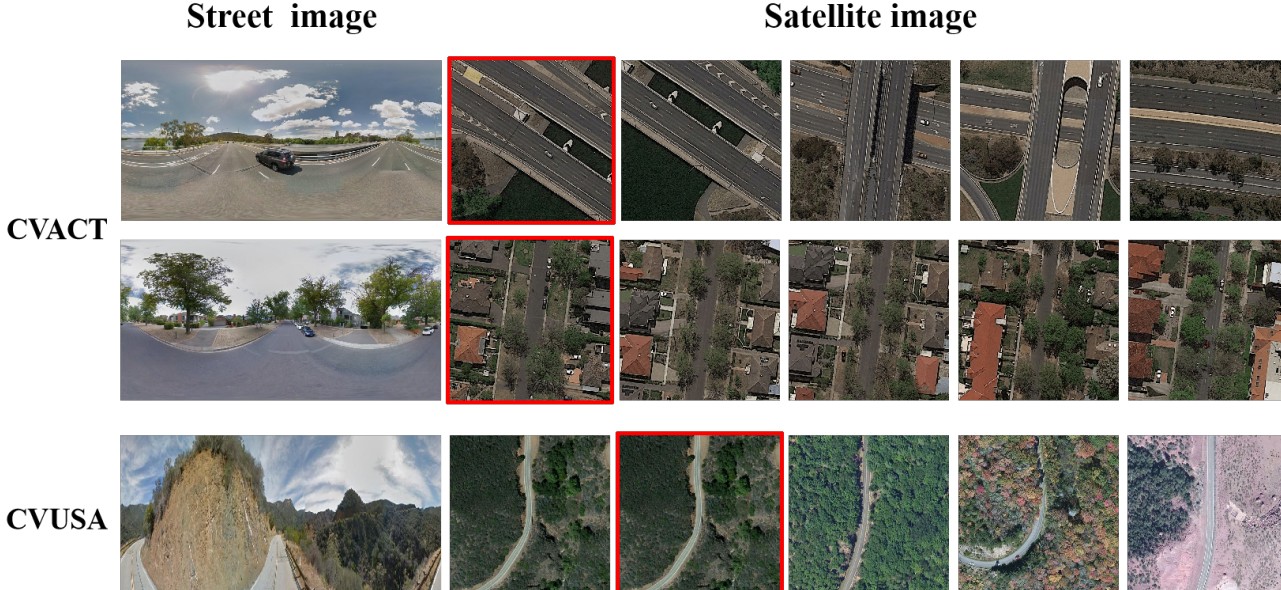

*Figure 7.* Retrieval visualizations on CVACT and CVUSA reveal that CVACT contains highly similar hard negatives, while CVUSA exhibits partially duplicated samples.

*Table 6.* Comparison of GFLOPs and average inference time per batch (batch size = 128) on CVACT_val. Avg inference time (ms) represents the mean time to process one batch.

| Method | GFLOPs | Avg Inference Time (ms) | CVACT_val R@1 |
|---|---|---|---|
| Sample4Geo | **90.47** | 2359.43 | 90.81 |
| Ours | 90.53 | **2356.45** | **92.87** |

## A. Semi-positive samples Visualization

We visualize retrieval results on CVACT and CVUSA and compute the local density to analyze dataset redundancy. As shown in Fig. 7, when using street images as queries and satellite images as references, CVACT contains highly similar top-2 neighbors, whereas CVUSA exhibits partially duplicated reference images. If such samples appear in the same batch and are treated as negatives, the model may overfit to noise, reducing generalization. Moreover, Fig. 8 shows that the density distribution of CVACT reference images is skewed to the right, indicating many visually similar scenes. Combined with the dense geographic sampling of CVACT, this suggests the existence of overlapping regions, semi-positive samples. This observation partially explains the significant generalization gains achieved by our method.

## B. Complexity Analysis

As shown in Table. 6, we report both GFLOPs and average inference time (in milliseconds per batch of 128 images) on the CVACT_val set. The results show that our method introduces only a negligible increase in parameters during training, while achieving even faster inference than the baseline. Notably, it still improves R@1 by nearly two points under performance-saturated settings, demonstrating a favorable accuracy–efficiency trade-off.

## C. Robustness Evaluation

To further verify that the performance gains stem from alleviating noise caused by over-separating semi-positive samples, we conduct additional experiments on a robustness-oriented dataset. Proposed in (Zhang & Zhu, 2024b), this dataset includes 16 types of corruptions to simulate realistic geo-localization conditions. As shown in Table. 7, even **without any explicit denoising modules or localization-aware data augmentation**, our method consistently outperforms approaches based on causal learning or heavy augmentation, with a clear and significant margin. These results strongly indicate that existing

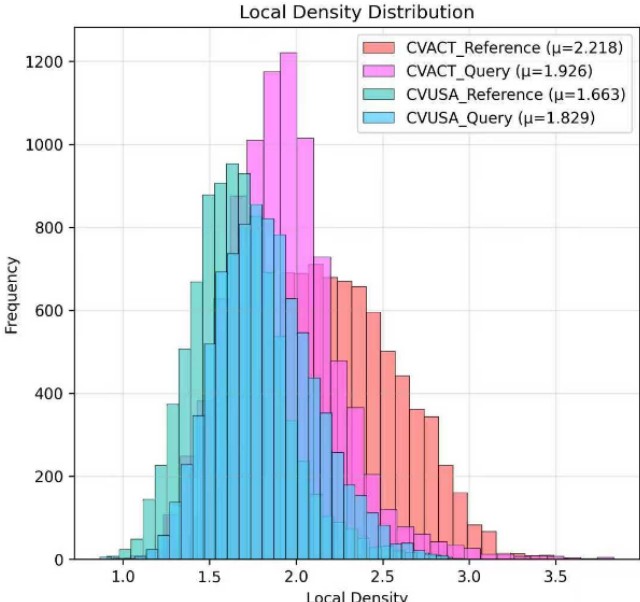

*Figure 8.* Local density distributions of CVACT and CVUSA, where a right-shift indicates the presence of a large number of similar samples.

*Table 7.* Comparisons with state-of-the-art models on the CVUSA-C-ALL, CVACT_val-C-ALL and CVACT_test-C-ALL datasets.

| Model | CVUSA-C-ALL | | | | CVACT_val-C-ALL | | | | CVACT_test-C-ALL | | | |
|---|---|---|---|---|---|---|---|---|---|---|---|---|
| | R@1 | R@5 | R@10 | R@1% | R@1 | R@5 | R@10 | R@1% | R@1 | R@5 | R@10 | R@1% |
| L2LTR | 87.93 | 95.45 | 97.01 | 99.01 | 82.13 | 93.34 | 94.93 | 98.10 | 57.20 | 82.59 | 87.23 | 98.09 |
| TransGeo | 82.72 | 91.95 | 94.03 | 97.92 | 74.04 | 86.19 | 89.10 | 94.98 | 52.18 | 74.35 | 78.99 | 95.03 |
| GeoDTR | 84.64 | 93.29 | 95.01 | 98.24 | 77.40 | 88.95 | 91.28 | 95.91 | 52.87 | 78.84 | 83.17 | 95.84 |
| EP-BEV | 86.22 | 94.86 | 96.58 | 99.00 | 85.94 | 94.52 | 95.93 | 98.21 | 64.62 | 87.75 | 90.78 | 98.43 |
| CLGT | 92.84 | 97.61 | 98.41 | 99.35 | 89.49 | 95.84 | 96.92 | 98.49 | 69.71 | 91.05 | 93.30 | **98.85** |
| Ours | **96.60** | **98.94** | **99.30** | **99.76** | **90.64** | **96.72** | **97.51** | **98.85** | **69.91** | **92.14** | **94.17** | 98.78 |

methods, which overlook the presence of semi-positive samples, tend to over-push such samples during training, leading to noise fitting and severely degraded generalization. In contrast, by modeling uncertainty with an NIG distribution and incorporating the proposed Soft InfoNCE loss, our method effectively mitigates this issue.

## D. GPU training time

As shown in Table 8, we further compare the training time between the baseline and our method on the CVACT dataset. Both methods are trained for 40 epochs using $4 \times$ 32GB NVIDIA V100 GPUs. The results show that our method achieves comparable training efficiency to the baseline, requiring approximately 5 hours of training time.

## E. Analysis of Top-K Sample Quality Across Training Epochs

To further address the issue where semi-positive samples and highly similar true negatives coexist, potentially leading to incorrect weakened repulsion, we conduct a statistical analysis on the CVACT dataset during the early-to-middle stages of training. Specifically, following the semi-positive definition in VIGOR, we set the similarity threshold to 0.4 and the GPS distance threshold to 25 m. The analysis is performed at intervals of 4 epochs during NIG training. For batches containing at least one semi-positive sample, we compute the frequency with which the Top-$K$ candidates contain highly similar true negative samples.

As shown in Table 9, even when semi-positive samples are present in Top-$K$, the frequency of highly similar true negative samples remains consistently low throughout training and further decreases as feature quality improves. These results

*Table 8.* Training time comparison on the CVACT dataset.

| Method | Hardware | Training Time |
|---|---|---|
| Baseline | $4 \times$ 32GB NVIDIA V100 GPUs | 5.1 hours |
| Ours | $4 \times$ 32GB NVIDIA V100 GPUs | 4.9 hours |

*Table 9.* Frequency of highly similar true negative samples in Top-$K$ when semi-positive samples are present, evaluated at 4-epoch intervals.

| Iteration | 1 | 2 | 3 | 4 | 5 |
|---|---|---|---|---|---|
| Frequency | 0.1139 | 0.0650 | 0.0430 | 0.0371 | 0.0321 |

*Table 10.* Ablation study on the regularization coefficient $\lambda_{reg}$ on the VIGOR dataset.

| $\lambda_{reg}$ | R@1 | R@5 | R@10 |
|---|---|---|---|
| 0.01 | **79.48** | 96.41 | 97.62 |
| 0.03 | 79.32 | **96.46** | **97.65** |
| 0.05 | 79.09 | 96.38 | 97.59 |

indicate that contamination from highly similar true negative samples is minimal across the early-to-middle training stages.

## F. Sensitivity Analysis of the Regularization Coefficient

We further evaluate the sensitivity of the regularization coefficient $\lambda_{reg}$ on the VIGOR dataset. Specifically, we test $\lambda_{reg} \in \{0.01, 0.03, 0.05\}$, and the results are reported in Table 10.

As shown in Table 10, the performance remains stable under different choices of $\lambda_{reg}$, indicating that the proposed method is relatively insensitive to this hyperparameter. This is mainly because $u$ only acts as a relative scaling factor for soft labels, rather than directly influencing predictions as in conventional DER-based methods. Although different $\lambda_{reg}$ values may affect the absolute magnitude of $u$, the normalization term $w$ maintains stable relative soft-label distributions, thereby ensuring robust performance.

