# OpenReview forum: "Must All Negatives Be Pushed Away Equally? Uncertainty-Aware Cross-View Geo-Localization via Normal Inverse Gamma Distribution"
_ICML.cc/2026/Conference — ICML 2026 regular_

### Official Review · Reviewer_bG2X · 2026-02-16

**Soundness:** 3
**Presentation:** 3
**Significance:** 3
**Originality:** 3
**Overall Recommendation:** 4
**Confidence:** 3

**Summary:**

This paper argues that a major reason cross-view geo-localization (matching a street image to the correct satellite image) fails to generalize to new cities is the way standard contrastive learning treats negatives. In many CVGL datasets, some negatives are actually semi-positives: they are geographically close to the true match and share strong semantic cues. Standard InfoNCE (even with label smoothing) pushes these hard negatives away too aggressively, which can make the model fit dataset-specific noise and later suffer a collapse in generalization.

Instead of pushing all negatives away equally, the model should adapt how strongly it repels hard negatives based on how ambiguous or complex the positive sample’s neighborhood is. The paper introduces uncertainty estimation to quantify this environmental complexity and uses it to soften labels for hard negatives.

The method models local neighborhood complexity via Deep Evidential Regression using a Normal-Inverse-Gamma (NIG) prior. It estimates a per-sample uncertainty value (computed in a single forward pass) and uses it to soften labels for the top-K hard negatives in a "Soft InfoNCE" objective. An uncertainty head based on CLS-to-spatial cross-attention, along with simple attention statistics, is proposed to fit the NIG parameters. The experiments claim strong performance and gains in zero-shot cross-dataset transfer, with an emphasized average R@1 improvement of 18%.

Overall, the manuscript’s major contribution is uncertainty-aware label adaptation for contrastive learning, aimed at reducing the over-separation of semi-positive negatives and improving cross-city generalization in CVGL. The study’s domain is cross-view geo-localization (street and satellite image retrieval).

**Compliance With Llm Reviewing Policy:**

Affirmed.

**Key Questions For Authors:**

- The method assumes Top-K most similar negatives are a good proxy for semi-positives. How robust is performance when Top-K includes truly wrong-but-similar negatives, and can you provide evidence (or analysis) that the selected Top-K predominantly correspond to geographically adjacent semi-positives throughout training (including early epochs)? Clarifying this could change my confidence in the method’s reliability across datasets.
- The approach introduces hyperparameters Top-K and w, and performance shifts across values. How should a practitioner choose these without dataset-specific tuning? Do you have a principled heuristic, sensitivity analysis across multiple datasets, or an adaptive strategy that reduces the need for tuning? A strong answer would improve my assessment.
- Please provide the proof (or derivation details) for Equation 9. If the derivation is standard, please cite a precise reference and include the key steps in an appendix. This affects technical completeness and reproducibility.
- Training time in GPU hour is missing.
- Several figures are hard to interpret (Figure 1 and Figure 2). Can you clarify the notation (e.g., P, E, the meaning of dashed lines and colors, and the meaning of highlighted colors in the attention map), and ensure all variables referenced in captions appear in the figure?

**Limitations:**

No. Please add a clearer limitations section that (i) discusses the reliance on Top-K as a proxy for semi-positives and when it might fail (ii) discusses hyperparameter sensitivity (Top-K, w) and the risk of dataset-specific tuning.

**Strengths And Weaknesses:**

Strengths:

- The paper is clearly written, and its claims are supported by experiments.
- Strong transfer results.
- Useful architectural details for the uncertainty head. The attention-statistics idea (entropy, max, top-2 ratio, variance) is simple and seems easy to implement.
- A concrete mechanism for label adaptation. The soft-label design ties increased negative probability mass to both similarity and estimated uncertainty for the top-K negatives, which directly targets the stated problem.
- Clear motivation linked to a real failure mode. The paper identifies a plausible and practically relevant issue: hard negatives that are actually semi-positive due to geographic proximity can create large gradients and drive representation learning toward noise fitting, harming generalization.
- Principled uncertainty modeling. Using an NIG conjugate prior and deriving uncertainty in closed form is conceptually neat and cheap at inference.

Weaknesses:
- It relies on top-K being a good proxy for “semi-positives”: The method selects the top-K most similar negatives and assumes these are the ambiguous ones worth softening. However, top-K can include truly incorrect negatives that happen to look similar, and softening them could hurt instance separation. In addition, the truly problematic negatives might not appear in the top-K early in training, which could hurt performance.
- Hyperparameters look important and non-universal: The approach introduces additional knobs, including top-K and the scaling factor w. The authors tune them and show performance shifts across values. This suggests that the method may require dataset-specific tuning (e.g., depending on how many semi-positives exist and how hard they are).
- Figure 1 is difficult to understand. The notation P and E is not clearly defined. These notations are introduced in the introduction, but notation B is missing where it states $1<i,j<B$. What do the two dashed lines associated with each $e_{ij}$ represent? What does the color of $e_{ij}$ indicate? In the later training stage (right figure), why do the two red $e_{ij}$ imply that the model fits the noisy parts of $e_{ij}$?
- Figure 2: The notation $s_{ij}$ is mentioned in the caption but not shown in the figure. The attention map contains two highlighted colors, but their meaning is not explained in the text.
- There is no clear explanation of why CLS attends to tokens in the CLS-to-spatial cross-attention module. The term spatial is also somewhat confusing, as it suggests geometry rather than content. I recommend using tokens instead of spatial.
- The proof for Equation 9 is not provided.
- Training time in GPU-hours is not reported.
- There is an irrelevant citation: "Langley, P. Crafting papers on machine learning. In Langley, P. (ed.), Proceedings of the 17th International Conference on Machine Learning (ICML 2000), pp. 1207–1216, Stanford, CA, 2000. Morgan Kaufmann"

Soundness:
Rating: 3 (good)
The central claims are reasonably supported by the reported experiments, especially the transfer results, and the method design is coherent with the stated failure mode (semi-positive hard negatives). The approach is technically plausible and lightweight at inference due to the one-pass uncertainty estimate. The paper has several soundness-adjacent gaps that limit confidence in full reproducibility and theoretical completeness, including the missing proof for Equation 9 and missing training-time reporting, as well as the method’s reliance on top-K as a proxy for semi-positives and sensitivity to top-K and w.

Presentation:
Rating: 3 (good)
The paper is generally clear and well motivated, and the architectural details are concrete enough to implement. However, multiple presentation issues materially hurt clarity in key places: Figure 1 is difficult to parse (notation and meaning of elements), and the "CLS-to-spatial cross-attention" wording is potentially confusing.

Significance:
Rating: 3 (good)
The paper targets a practically relevant and widely observed weakness in CVGL: poor generalization to new cities and datasets. The proposed uncertainty-aware softening of hard negatives is a concrete intervention on a common contrastive learning failure mode, and the strong transfer improvements suggest potential real impact for robust deployment. While the scope is specialized to CVGL, the framing could generalize to other retrieval settings where semi-positives exist.

Originality:
Rating: 3 (good)
Originality mainly comes from applying deep evidential regression with an NIG prior to estimate per-sample neighborhood complexity, then using that uncertainty to adapt the label mass assigned to hard negatives in a Soft InfoNCE objective. The uncertainty head design using attention statistics is also a simple, implementable contribution. The novelty is less about inventing entirely new components and more about a creative, well-motivated integration to address a specific generalization failure mode.

---

> ### Author Rebuttal · Authors · 2026-03-29
>
> ## Response to Reviewer 4 (bG2X)
>
> We thank **Reviewer bG2X** for their insightful remarks and helpful suggestions. The feedback has played an important role in improving the overall quality of the manuscript. We have addressed all the concerns raised and revised the manuscript accordingly. The key changes are summarized as follows:
>
> ### Q1. Regarding Top-K selection and training stability in early epochs
>
> **A1.** We sincerely thank the reviewer for this thoughtful question. We would like to kindly clarify the Top-K selection has principled dataset-level support: VIGOR officially annotates exactly 3 semi-positive samples per query, so K=5 covers all semi-positives with only a small buffer, giving it clear semantic justification. For CVACT/CVUSA, K=5 is similarly supported by neighbor density estimation.
>
> Regarding wrong-but-similar negatives, we address this from three angles. First, we adopt Sample4Geo's similarity-based and GPS-based sampling [1], ensuring Top-K candidates effectively cover genuine semi-positives. Second, the 4-epoch encoder warmup with the Head frozen ensures basic feature discriminability before Head training begins, making early Top-K candidates predominantly semantically similar semi-positives rather than random confusers. Third, even if a few true negatives enter Top-K, the NIG distribution models the overall statistical properties of $t_{ij}$ rather than individual samples, making $u_i$ robust to occasional noise.
>
> [1] Deuser et al. Sample4Geo, ICCV 2023.
>
> ### Q2. Regarding the selection of $w$ and hyperparameter tuning
>
> **A2.** We appreciate the reviewer raising this concern. We clarify the choice of $w$ follows a principled derivation in Appendix E based on gradient suppression ratio $\rho$:
>
> $$w = \frac{(1-\rho)(p_{hard} - \lambda_{base})}{s_{hard} \cdot 0.5}$$
>
> where $s_{hard}$ is simply the average maximum Top-K similarity over the training set — requiring no annotation. The 40%–50% suppression target is principled: below 30% risks collapsing semi-positive labels toward positive labels, breaking discriminative boundaries; above 70% approximates standard InfoNCE, failing to alleviate over-repulsion. Therefore, targeting ~50% gradient compression for hard negatives, we derive $w$ from the formulation, rather than relying on dataset-specific tuning; details are provided in Appendix Sec. E.
>
> ### Q3 & Q4. Regarding the description of Fig. 1 and notation in Fig. 2
>
> **A3&4.** We sincerely thank the reviewer for pointing out these clarity issues and will strengthen the explanation in the revised manuscript. Briefly: $e_{ij}$ represents image pair information — green denotes localization-relevant content, red denotes noise. $P$ and $E$ denote the localization capability acquired during training and required for correct evaluation respectively, with dashed arrows indicating $P$'s expansion direction. In late training, over-separation of semi-positives forces $P$ to expand into noisy regions of $e_{ij}$, stagnating $G=P\cap E$. For the attention map, warm and cool colors indicate high- and low-weight regions of the cls-to-spatial attention respectively.
>
> ### Q5. Regarding the derivation of Eq. 9
>
> **A5.** Eq. 9 is the negative log marginal likelihood under a Gaussian likelihood with an NIG conjugate prior. Given observation $t_{ij} \sim \mathcal{N}(\mu_i, \sigma_i^2)$, we place an NIG prior over $(\mu_i, \sigma_i^2)$ (Eq. 3) and integrate out these parameters:
>
> $$p(t_{ij} \mid \gamma_i, \nu_i, \alpha_i, \beta_i) = \int\int \mathcal{N}(t_{ij};\mu_i,\sigma_i^2) \cdot p(\mu_i,\sigma_i^2), d\mu_i, d\sigma_i^2$$
>
> By Gaussian-NIG conjugacy, this yields a closed-form Student-t distribution (Amini et al., NeurIPS 2020). Taking its negative log and letting $\Omega_i = 2\beta_i(1+\nu_i)$ directly gives Eq. 9:
>
> $$\mathcal{L}^i_{NLL} = \frac{1}{2}\log\frac{\pi}{\nu_i} - \alpha_i \log\Omega_i + \left(\alpha_i + \frac{1}{2}\right)\log\left[(t_{ij}-\gamma_i)^2\nu_i + \Omega_i\right] + \log\Gamma(\alpha_i) - \log\Gamma\left(\alpha_i+\frac{1}{2}\right)$$
>
> Full derivation details are provided in Amini et al., "Deep Evidential Regression," NeurIPS 2020.
>
> ### Q6. Regarding GPU training time
>
> **A6.** We thank the reviewer for raising this. We will include training time details in the revised manuscript. Taking the CVACT dataset as an example, our method is trained for 40 epochs on 4 × 32GB NVIDIA V100 GPUs, requiring approximately 5 hours — on par with the baseline, as shown in Table I.
>
> **Table I.** Training time comparison.
>
> | Method   | Hardware                  | Training Time |
> | -------- | ------------------------- | ------------- |
> | Baseline | 4 × 32GB NVIDIA V100 GPUs | 5.1 hours     |
> | Ours     | 4 × 32GB NVIDIA V100 GPUs | 4.9 hours     |
>
> ### Q7. Regarding writing and citation issues
>
> **A7.** We sincerely thank the reviewer for pointing out the issues with the attention module description and related citations. We will correct any potential errors and ensure accuracy in the final version.

---

> > ### Author Rebuttal · Reviewer_bG2X · 2026-04-01
> >
> > Thank you for the detailed rebuttal. My concerns are partially resolved.
> >
> > The rebuttal usefully addresses several points. In particular, the added explanation for Eq. 9 as the negative log marginal likelihood under a Gaussian likelihood with an NIG prior is helpful, and citing DER / Amini et al. clarifies where the derivation comes from. The added training-time information is also useful, and the clarifications for Fig. 1 and Fig. 2 suggest that the presentation issues can likely be improved in the revision. I also appreciate the commitment to fix the wording around the attention module and remove citation issues.
> >
> > My main technical concern is only partially addressed. The core claim of the method is that the Top-$K$ selected negatives are a reliable proxy for semi-positives. The rebuttal provides a plausible rationale based on VIGOR annotations, neighbor density in CVACT/CVUSA, Sample4Geo sampling, and a 4-epoch warmup. However, from the rebuttal alone, I still do not see direct evidence that the selected Top-$K$ negatives are indeed predominantly geographically adjacent or semi-positive throughout training, especially in early and middle epochs. This matters because if a nontrivial fraction of Top-$K$ entries are truly incorrect but visually similar negatives, then softening them may weaken instance discrimination.
> >
> > Similarly, the explanation for the hyperparameter $w$ is more principled than before, but I still view this point as only partially resolved. The gradient-suppression-ratio argument is interesting, but based on the rebuttal alone, it remains unclear how robust this heuristic is across datasets without further tuning.
> >
> > Overall, the rebuttal improves my confidence somewhat, but not enough to change my scores at this stage.

---

> > > ### Author Response · Authors · 2026-04-04
> > >
> > > We sincerely thank the reviewer for the further attention and are glad the rebuttal partially addressed your concerns. We apologize that character limits prevented a more complete response on this point in first rebuttal, and provide additional clarification here.
> > >
> > > Regarding the question of whether the negative samples in Top-$K$ may be semi-positive samples, we wish to highlight a key design property: thanks to the sampling strategy and warmup mechanism, our NIG model learns uncertainty information from the positive samples. When highly similar true negative samples appear in the Top-K of a clean sample, the NIG predicts low uncertainty $u$, resulting in smaller soft labels and stronger push-away for these samples, exactly as desired. Conversely, when the positive resides in a dense semi-positive neighborhood, NIG outputs high $u$. This self-regulating property means the method naturally handles both cases without manual intervention, which is also evidenced by the clear performance gap over the similarity-weighted baseline (ablation without uncertainty, using only similarity weighting) shown in Table II.
> > >
> > > **Table II.** Ablation on cross-dataset transfer (CVACT $\rightarrow$ CVUSA).
> > >
> > > | Setting             | R@1       | R@5       | R@10      |
> > > | ------------------- | --------- | --------- | --------- |
> > > | Similarity-weighted | 60.56     | 77.57     | 83.17     |
> > > | Ours                | **69.34** | **83.39** | **96.89** |
> > >
> > > To further address the issue where semi-positive samples and highly similar true negatives coexist, leading to incorrect weakened repulsion, we conduct a statistical analysis on CVACT for the early-to-middle training stages. Specifically,  we set a similarity threshold of 0.4 and a GPS distance threshold of 25m (following VIGOR's semi-positive definition), and conduct the analysis at intervals of 4 epochs when NIG is trained. For batches containing at least one semi-positive sample, we then compute the frequency with which the Top-𝐾 candidates contain highly similar true negative samples.
> > >
> > > **Table III.** Frequency of highly similar true negative samples in Top-K when semi-positive samples are present, evaluated at 4-epoch intervals.
> > >
> > > | ITERATION | 1      | 2      | 3      | 4      | 5      |
> > > | --------- | ------ | ------ | ------ | ------ | ------ |
> > > | Frequency | 0.1139 | 0.0650 | 0.0430 | 0.0371 | 0.0321 |
> > >
> > > As shown in Table III,  even when semi-positives are present in Top-K, the frequency of highly similar true negative samples remains consistently low throughout training and decreases further as feature quality improves, confirming that contamination from highly similar true negative samples is minimal across the early-to-middle training stages.
> > >
> > > Regarding $w$, the principled derivation in Appendix E greatly reduces hyperparameter tuning cost — no additional annotation or validation set tuning is needed. We hope these clarifications fully address your remaining concerns and thank you sincerely for your time and constructive feedback.

---

### Official Review · Reviewer_2NjJ · 2026-03-03

**Soundness:** 2
**Presentation:** 3
**Significance:** 2
**Originality:** 1
**Overall Recommendation:** 3
**Confidence:** 4

**Summary:**

This paper addresses cross-view geo-localization (CVGL). The authors argue that a key failure mode of contrastive-learning-based CVGL models is overly large gradients from semi-positive samples, which can cause the model to overfit to spurious/noisy signals rather than informative cues. To mitigate this, they propose soft-InfoNCE, which reweights the InfoNCE loss using uncertainty estimated via Deep Evidential Regression (DER), thereby down-weighting semi-positive pairs with high uncertainty. Experiments on multiple datasets suggest improved performance and better generalization compared to prior work.

**Compliance With Llm Reviewing Policy:**

Affirmed.

**Final Justification:**

I mainly raised concerns about (1) how epistemic uncertainty obtained from DER is applied and (2) the similarity of the proposed contrastive learning method to [1].

1. First, it appears that the authors do not fully distinguish between epistemic and aleatoric uncertainty. In the rebuttal, the authors initially stated that epistemic uncertainty is not used in this work. Although the authors later acknowledged that this statement was incorrect, the subsequent explanation remains difficult to reconcile with the formulation of DER. For example:

> The Var[$\mu$] term serves primarily as a correction for mean estimation error under limited Top-K observations, rather than as an independent epistemic contribution.

> The benefit of DER lies in estimation robustness under limited observations, not in distinguishing uncertainty types.

2. The authors argue that the motivation for DER-based contrastive learning differs from [1], in that this work aims to reduce the gradient contribution of semi-positive samples. However, a similar effect could likely be achieved in [1] by adjusting the regularization coefficient, since evidential learning methods are known to be highly sensitive to this coefficient [2–5]. Even if the authors’ formulation is technically correct, the additional mechanism for regulating gradients appears to provide only marginal novelty relative to prior work. The conceptual contribution seems incremental rather than fundamentally distinct from existing evidential contrastive learning formulations.

In conclusion, based on the above concerns, I would like to maintain my score at weak reject.

References
* [1] Wang, Yifan, et al. Towards Robust Uncertainty Calibration for Composed Image Retrieval. NeurIPS 2025.
* [2] Shen, Maohao, et al. Are Uncertainty Quantification Capabilities of Evidential Deep Learning a Mirage? NeurIPS 2024.
* [3] Meinert, Nis, Jakob Gawlikowski, and Alexander Lavin. The Unreasonable Effectiveness of Deep Evidential Regression. AAAI 2023.
* [4] Juergens, Mira, et al. Is Epistemic Uncertainty Faithfully Represented by Evidential Deep Learning Methods? arXiv:2402.09056, 2024.
* [5] Amini, Alexander, et al. Deep Evidential Regression. NeurIPS 2020.

**Key Questions For Authors:**

**Questions related to weakness #3**
1. Several studies [2,3,4] argue that evidential learning methods do not reliably represent epistemic uncertainty. How do you justify using DER-based uncertainty as a proxy for epistemic uncertainty in your setting?
2. Evidential learning is known to be sensitive to the regularization coefficient (e.g., ​\lambda_{reg} in Eq. (11)) [2,4,5]. Did you perform a sensitivity analysis over \lambda_{reg}, and how robust are your results to this choice?

* References

[2] Shen, Maohao, et al. “Are uncertainty quantification capabilities of evidential deep learning a mirage?” NeurIPS 2024.

[3] Meinert, Nis, Jakob Gawlikowski, and Alexander Lavin. “The unreasonable effectiveness of deep evidential regression.” AAAI 2023.

[4] Juergens, Mira, et al. “Is epistemic uncertainty faithfully represented by evidential deep learning methods?” arXiv:2402.09056 (2024).

[5] Amini, Alexander, et al. “Deep evidential regression.” NeurIPS 2020.

**Limitations:**

1. **(Weakness #1)** The core motivation and formulation appear very close to [1]. Both works employ an uncertainty-weighted contrastive objective where uncertainty from DER is used to reweight pairwise contributions. While [1] studies composed image retrieval (CIR) and this paper studies CVGL, both are fundamentally retrieval problems in different domains/modality settings. I acknowledge that the authors’ novelty may come from modality-specific architectural choices and weighting details (e.g., [1] uses total uncertainty as the weight), but the paper would benefit from additional evidence to establish distinct contributions beyond transferring the same idea to CVGL (e.g., stronger ablations, justification of weighting design, or modality-specific analysis).
2. **(Weakness #2)** In [1], the authors attribute aleatoric uncertainty to semantic ambiguity and annotation noise, while epistemic uncertainty is tied to model overconfidence in dominant categories. Based on [1] and the current manuscript, the dominant uncertainty in CVGL may also be largely aleatoric. If so, a simpler uncertainty model (e.g., variance from a Gaussian NLL) might already be sufficient. This point is also linked to the justification of using evidential learning for epistemic uncertainty (Key Questions). The paper should more thoroughly analyze what drives uncertainty in CVGL and why DER is the appropriate tool.
3. (minor) The justification that t_ij follows a Gaussian distribution is not fully convincing. Since cosine similarity lies on a spherical manifold, a von Mises-Fisher-type distribution may be more appropriate. It would be more technically accurate to state that the similarity distribution is approximated by a Gaussian under certain assumptions.
4. (minor)  Some notations appear to refer to the same quantity:
    - t_ij and s_ij in Eqs. (6) and (8) seem to represent the same similarity term.
    - \lambda^soft_ij and y^soft_ij appear to be defined as \lambda_base + s_ij * u_ij * w. Please clarify the definitions and ensure consistent notation.

**Strengths And Weaknesses:**

*  **Strengths**
1. The paper is clearly written, and the proposed hypothesis about the error source in CVGL, as well as the overall methodology, is easy to follow.
2.The experiments demonstrate strong performance (including state-of-the-art results) and generalization across multiple datasets compared to prior methods.

* **Weaknesses**
1. The proposed approach appears highly similar to prior work in a related image-retrieval setting [1], particularly in terms of the loss formulation and the core idea of uncertainty-weighted contrastive learning.
2. The paper does not clearly characterize the source of uncertainty in CVGL—i.e., whether it is primarily aleatoric, epistemic, or a mixture. This distinction matters because the appropriate uncertainty quantification strategy depends on the dominant uncertainty source.
3. The claim that evidential learning (DER/EDL-style methods) can reliably quantify epistemic uncertainty is not sufficiently validated.

* References

[1] Wang, Yifan, et al. “Towards Robust Uncertainty Calibration for Composed Image Retrieval.” NeurIPS 2025.

---

> ### Author Rebuttal · Authors · 2026-03-29
>
> ## Response to Reviewer 3 (2NjJ)
>
> We are grateful to **Reviewer 2NjJ** for their valuable feedback and detailed evaluation. The suggestions have been very helpful in strengthening our work.  The major updates are summarized as follows:
>
> ### Q1. Regarding originality clarification
>
> **A1.** We sincerely thank the reviewer for raising this point. We would like to kindly clarify two aspects.
>
> First, although both works adopt DER, their core motivations differ. [1] focuses on CIR, modeling NIG over all candidates to amplify gradients and prevent underfitting. In contrast, our method targets CVGL, models NIG over Top-K hard negatives to estimate neighborhood complexity $u_i$, and applies Soft InfoNCE to suppress over-repulsion on semi-positives, effectively reducing gradients. The two designs address **opposite issues**: [1] targets underfitting, while ours focuses on overfitting, each arising from distinct task challenges. Furthermore, Appendix E provides a gradient-level derivation showing that our method compresses hard negative gradients by ~50%, while the ablations (Response to Reviewer 2, Table II) confirm that the gains come from our NIG-based design rather than generic reweighting, supporting the originality of our contribution.
>
> Second, since [1] was not on arXiv, we only came across it near the ICML 2026 submission deadline, when our work was nearly complete; thus, the two works are independent concurrent contributions.
>
> ### Q2. Regarding epistemic uncertainty and regularization coefficient
>
> #### Q2-1. Epistemic uncertainty
>
> **A2-1.** We sincerely thank the reviewer for this insightful point. We will revise the description of epistemic uncertainty in the final version. In the CVGL setting,  $u$ is predominantly aleatoric — semi-positive samples represent intrinsic data-level ambiguity from dense geographic sampling. We therefore agree with the reviewer that the dominant uncertainty here is largely aleatoric, consistent with [1]. The $\text{Var}[\mu]$ term serves primarily as a correction for mean estimation error under limited Top-K observations, rather than as an independent epistemic contribution. As suggested, we conducted the following ablation experiments on VIGOR:
>
> **Table I.** Ablation study on Gaussian NLL variance and aleatoric uncertainty.
>
> | Variant               | R@1       | R@5       | R@10      |
> | --------------------- | --------- | --------- | --------- |
> | Gaussian NLL variance | 78.31     | 95.87     | 97.22     |
> | E[σ²] only            | 78.89     | 96.15     | 97.41     |
> | E[σ²] + Var[μ] (Ours) | **79.48** | **96.41** | **97.62** |
>
> The training loss for the Gaussian NLL variant is:
>
> $$L_{Gauss} = \frac{1}{B \cdot K}\sum_i\sum_j \left[\frac{(t_{ij}-\hat{\mu}_i)^2}{2\hat{\sigma}^2_i} + \frac{1}{2}\log\hat{\sigma}^2_i\right]$$
>
> Here $\hat{\sigma}^2_i$ directly replaces $u_i$ in the soft label formula (Eq. 8), and the head outputs two parameters.
>
> Regarding the justification for DER over simpler alternatives, our choice does not rely on epistemic uncertainty at all, but on two practical points. First, Gaussian NLL outputs a fixed variance estimate with no notion of its own reliability; NIG instead models a distribution over the variance itself, yielding more robust estimates under limited Top-K=5 observations. Second, as shown in Table I, even when aleatoric uncertainty dominates, NIG's higher-order modeling provides consistent gains over both Gaussian NLL variance and E[$\sigma^2$]-only. The benefit of DER lies in estimation robustness under limited observations, not in distinguishing uncertainty types.
>
> #### Q2-2. Regularization coefficient
>
> **A2-2.**  We sincerely thank the reviewer for raising this important point. As suggested, we evaluated $\lambda_{reg} \in {0.01, 0.03, 0.05}$ on VIGOR (Table II) and found our method relatively insensitive to this choice. The key reason is that $u$ serves only as a relative scaling factor for soft labels rather than directly affecting predictions as in standard DER. Even if $\lambda_{reg}$ shifts the absolute value of $u$, the normalization provided by $w$ keeps relative soft label magnitudes stable, ensuring robustness to such changes.
>
> **Table II.** Ablation study on the $\lambda_{reg}$.
>
> | λ_reg | R@1       | R@5       | R@10      |
> | ----- | --------- | --------- | --------- |
> | 0.01  | **79.48** | 96.41     | 97.62     |
> | 0.03  | 79.32     | **96.46** | **97.65** |
> | 0.05  | 79.09     | 96.38     | 97.59     |
>
> ### Q3. Regarding the Gaussian distribution assumption
>
> **A3.** We genuinely appreciate the reviewer's technically precise observation. Please see our response to Reviewer 1 Q3. We will revise the description to acknowledge this as a Gaussian approximation for greater rigor.
>
> ### Q4. Regarding variable naming issues
>
> **A4.** We sincerely thank the reviewer for pointing out the variable naming details. We will carefully review and correct these in the revised version to ensure consistency and clarity throughout the manuscript.

---

> > ### Author Rebuttal · Reviewer_2NjJ · 2026-04-02
> >
> > Thank you for the detailed rebuttal.
> >
> > ### Q1.
> > The authors state that
> > > [1] focuses on CIR, modeling NIG over all candidates to amplify gradients and prevent underfitting.
> >
> > Based on my understanding of [1], the gradient magnitude is reduced when the total uncertainty u_i is smaller than 1.0, and increases otherwise. Therefore, this statement is partly not correct. Although I acknowledge that the authors designed the method to reduce repulsion among the top-k samples, I believe the fundamental contribution lies in feature alignment via uncertainty-weighted contrastive learning.
> >
> > ### Q2-1.
> > I believe the authors are confused about the concept of epistemic uncertainty. In the rebuttal, the authors argue that
> >
> > > Regarding the justification for DER over simpler alternatives, our choice does not rely on epistemic uncertainty at all, but on two practical points
> >
> > However, DER states that Var[$\mu$] represents epistemic uncertainty. It is therefore unclear how the authors can adopt DER while simultaneously claiming that their approach does not rely on epistemic uncertainty.
> >
> > Moreover, the following statement appears inconsistent with the authors’ earlier arguments:
> >
> > > our choice does not rely on epistemic uncertainty at all, but on two practical points. First, Gaussian NLL outputs a fixed variance estimate with no notion of its own reliability; NIG instead models a distribution over the variance itself
> >
> > In an ideal setting with noisy data, the predicted variance distribution should concentrate around a single value as more observations are collected. The distribution over the variance is directly affected by the amount of available data, which is precisely the source of epistemic uncertainty.
> >
> > > The benefit of DER lies in estimation robustness under limited observations, not in distinguishing uncertainty types.
> >
> > As noted above, limited observations, i.e., lack of data, are exactly what give rise to epistemic uncertainty.
> >
> > ### Q2-2.
> >
> > I appreciate the experiments evaluating different regularization constants. However, I believe a wider range of λreg\lambda_{\text{reg}}λreg​ should be considered. For example, [1] and [2] perform sensitivity analyses over the ranges [0, 0.01, …, 0.2] and [1e-3, 1e-2, 1e-1], respectively.
> >
> > ### Q3 & Q4.
> > Thank you for addressing the concerns related to these questions.
> >
> > Overall, I would like to maintain my current scores, as my concerns have not been fully resolved.

---

> > > ### Author Response · Authors · 2026-04-04
> > >
> > > We sincerely thank Reviewer 2NjJ for the careful reading and the accurate corrections to our first rebuttal. The compressed character limit in our initial response unfortunately led to ambiguous phrasing and inaccurate characterizations, for which we offer our sincere apology. We are grateful for the opportunity to clarify these points more precisely below.
> > >
> > > ## Response to Q1
> > >
> > > We thank the reviewer for pointing this out. From [1] (Section 3.2, Eq. 8), the uncertainty-weighted loss is:
> > > $L_{\text{bc}} = -\frac{1}{B} \sum_{i=1}^B u_i \cdot \log \frac{\exp(s_{ii}/\tau)}{\sum_j \exp(s_{ij}/\tau)}$.
> > > [1] explicitly motivates this design by stating: "When a semantic embodies high uncertainty, the corresponding weight is ought to set larger... avoiding neglecting infrequent semantic categories." As confirmed by the original context, [1] assigns larger weights (i.e., larger gradients) to positive samples with high uncertainty semantics, so that the model continues learning these under-learned categories and prevents under-learning caused by semantic distribution imbalance. We appreciate the reviewer's careful correction — since not all samples satisfy $u_i > 1.0$, this does not contradict the reviewer's observation that gradients decrease when $u_i < 1.0$. The key point is that [1]'s core design objective is to increase gradients for under-learned positive samples. In contrast, we compress gradients for semi-positives to prevent generalization degradation caused by excessive repulsion, thus our core motivation and method are fundamentally different. While both use uncertainty-weighted contrastive learning as a starting point, our identification of the generalization collapse problem specific to CVGL and this task-specific gradient compression design constitute a distinct contribution beyond the general idea.
> > >
> > > ## Response to Q2-1
> > >
> > > We sincerely thank the reviewer for this conceptual clarification and apologize for the wording error in our first rebuttal: due to character compression, we mistakenly wrote **"does not rely on epistemic uncertainty at all"** when we actually meant **"does not rely solely on epistemic uncertainty"**. This inaccurate phrasing led to the reviewer's reasonable questioning, and we deeply regret this mistake. As we stated in our first rebuttal (A2-1), "The Var[μ] term serves primarily as a correction for mean estimation error under limited Top-K observations, rather than as an independent epistemic contribution," which makes clear that both aleatoric and epistemic components work together in our method. We fully agree with the reviewer on the DER/NIG framework conceptual perspective. This is exactly why we adopt DER — we need the richer modeling that DER provides under limited observations. Our ablation study in Table I and subsequent experimental analysis demonstrate that we benefit from both types of uncertainty. We will revise the wording in the final version to avoid any ambiguity.
> > >
> > > ## Response to Q2-2
> > >
> > > We sincerely thank the reviewer for this suggestion. Following the reviewer's recommendation, we extended the sensitivity analysis to a wider range of $\lambda_{reg}$ (Table II). Performance remains stable in the lower range and degrades gracefully only when $\lambda_{reg}$ becomes sufficiently large to fundamentally distort the uncertainty scale. Specifically, excessive $\lambda_{reg}$ over-suppresses evidence accumulation, causing $u = \frac{\beta(\nu+1)}{\nu(\alpha-1)}$ to be driven uniformly high by inflating $\beta$ and suppressing $\nu, \alpha$, which over-softens negative labels and weakens discriminability. Crucially, as established in our first rebuttal, the normalization provided by $w$ buffers moderate $\lambda_{reg}$ shifts on relative soft label magnitudes, explaining the observed robustness within the stable region.
> > >
> > > **Table II.** Ablation study on the $\lambda_{reg}$.
> > >
> > > | λ_reg | R@1   | R@5   | R@10  |
> > > | ----- | ----- | ----- | ----- |
> > > | 0.001 | 79.22 | 96.29 | 97.53 |
> > > | 0.01  | 79.48 | 96.41 | 97.62 |
> > > | 0.03  | 79.32 | 96.46 | 97.65 |
> > > | 0.05  | 79.09 | 96.38 | 97.59 |
> > > | 0.10  | 78.57 | 96.13 | 97.35 |
> > > | 0.15  | 78.02 | 95.84 | 97.03 |
> > > | 0.20  | 77.47 | 95.54 | 96.68 |

---

### Official Review · Reviewer_uyC3 · 2026-03-04

**Soundness:** 2
**Presentation:** 3
**Significance:** 2
**Originality:** 2
**Overall Recommendation:** 4
**Confidence:** 4

**Summary:**

This paper proposes an uncertainty-aware contrastive learning framework for cross-view geo-localization (CVGL). The authors argue that standard InfoNCE assigns equal labels to all negatives, which can lead to over-separation of semi-positive samples and degrade generalization in unseen environments. To address this issue, they introduce a Deep Evidential Regression (DER) framework that models the similarity distribution between a positive sample and its Top-K hard negatives using a Normal-Inverse-Gamma (NIG) prior. The estimated uncertainty is then incorporated into a Soft InfoNCE objective to adaptively soften repulsion for hard negatives.

**Compliance With Llm Reviewing Policy:**

Affirmed.

**Final Justification:**

The authors provided targeted responses to my main concerns, particularly by adding comparisons with recent methods and conducting controlled experiments to disentangle the effects of NIG-based uncertainty modeling from more general negative reweighting/softening. These additions strengthen the paper’s experimental support.

**Key Questions For Authors:**

1. The empirical evidence does not fully substantiate the central claims.The paper argues that uncertainty-aware softening mitigates generalization collapse caused by equal-negative treatment. Yet, improvements on VIGOR are relatively modest.

2. The experimental evaluation also omits several recent high-performing CVGL methods (e.g., Panorama-BEV (ECCV 2024) and AuxGeo (ISPRS 2025)), which report stronger performance on VIGOR. This weakens the claim that the proposed mechanism fundamentally addresses the issue, making it difficult to determine whether the method truly advances the state of the art.

3. In addition, the paper does not provide formal generalization guarantees or controlled experiments that isolate the effect of uncertainty modeling from generic negative reweighting. As a result, it remains unclear whether the observed improvements stem specifically from the proposed NIG-based uncertainty estimation or from a general softening of negative labels.

**Limitations:**

See weaknesses above.

**Strengths And Weaknesses:**

- Strengths
1. Overall, the manuscript's major contribution pertains to re-examining equal-negative contrastive learning under semi-positive ambiguity in cross-view geo-localization. The set-theoretic interpretation of generalization (P ∩ E) provides an intuitive narrative for understanding overfitting caused by excessive separation of hard negatives. This conceptual framing is well articulated and offers a useful perspective on robustness in retrieval-based systems.
2. The integration of Deep Evidential Regression with a Normal-Inverse-Gamma prior is mathematically valid and aligns with prior literature. The uncertainty is computed in a single forward pass, which is computationally efficient compared to Monte Carlo approaches. The loss functions are clearly defined, and gradient behavior is explicitly analyzed. Overall, the methodological design is technically sound.
3. The reported improvements in cross-dataset transfer are substantial. If validated, these results suggest that uncertainty-aware negative softening can improve robustness in unseen environments, which is practically relevant for geo-localization tasks.
4. The method introduces minimal computational overhead and includes ablation studies for Top-K and scaling parameters. The complexity analysis and Gaussian modeling validation strengthen the empirical thoroughness.
- Weaknesses
1. The method combines Deep Evidential Regression, Normal-Inverse-Gamma uncertainty modeling, and a softened InfoNCE loss. While the integration is thoughtful, the individual components are established techniques. The contribution is primarily a principled adaptation and combination tailored to CVGL, rather than a fundamentally new theoretical development. The originality is therefore solid but not groundbreaking.
2. Although the paper critiques equal-negative contrastive learning, it does not compare against other false-negative mitigation or confidence-aware contrastive strategies outside the CVGL literature. The proposed soft labeling mechanism may be conceptually related to adaptive reweighting or debiased contrastive approaches, yet no empirical comparison is provided. Without such comparisons, it is unclear whether the observed gains arise specifically from NIG-based uncertainty modeling or from a more general negative reweighting effect.
3. Despite the paper reports strong results against several prior CVGL methods, recent high-performing approaches such as Panorama-BEV (ECCV 2024) and AuxGeo (ISPRS 2025) achieve stronger performance on key benchmarks according to their published results. These methods are not included in the comparisons based on VIGOR dataset. Without direct comparison, it is difficult to assess whether the proposed approach truly advances the state of the art or is competitive only against a subset of prior work.
4. The paper provides strong intuition for why uncertainty-weighted soft labels improve generalization. However, no formal generalization bounds or theoretical guarantees are provided to support the set-theoretic argument. The theoretical contribution remains explanatory rather than formally predictive.

---

> ### Author Rebuttal · Authors · 2026-03-29
>
> ## Response to Reviewer 2 (uyC3)
>
> We thank **Reviewer uyC3** for their thorough review and constructive feedback. The comments have greatly contributed to enhancing the overall quality of the paper. The main updates are summarized as follows:
>
> ### Q1. Regarding performance on VIGOR
>
> **A1.** We thank the reviewer for this observation. As noted in Section 4.2 (line 345), the limited gains on VIGOR stem from a dataset-structural constraint: the cross-modal similarity gap between semi-positives and ordinary negatives in VIGOR is inherently small (~0.2), compared to ~0.6 in CVACT. This physically limits the discriminability that soft labels can provide. Nevertheless, even under this challenging setting, our method still achieves consistent gains (Same +2%, Cross +5%). Notably, the 5% improvement on VIGOR Cross-area, which directly evaluates cross-city generalization, validates the effectiveness of our approach on the paper's core claim.
>
> ### Q2. Regarding comparison with Panorama-BEV and AuxGeo
>
> **A2.** We sincerely thank the reviewer for raising these comparisons. We clarify that EP-BEV in Tables 1–3 corresponds to Panorama-BEV (Ye et al., ECCV 2024), evaluated under the same image size setting as ours (following CLGT, NeurIPS 2025), and is therefore already included in all comparisons, where our method outperforms it in both performance and efficiency. For AuxGeo, our method achieves better results on the more challenging cross-area setting (Table I), which directly evaluates generalization and is the core focus of our work, while remaining competitive on same-area. Notably, both methods rely on additional BEV modality whereas ours uses only standard street-satellite pairs.
>
> **Table I.** Comparison with AuxGeo on the VIGOR dataset.
>
> | Method | Same-area |           |           | Cross-area |           |           |
> | ------ | --------- | --------- | --------- | ---------- | --------- | --------- |
> |        | R@1       | R@5       | R@10      | R@1        | R@5       | R@10      |
> | AuxGeo | **80.34** | 96.25     | 97.57     | 63.94      | 84.98     | 88.98     |
> | Ours   | 79.48     | **96.41** | **97.62** | **66.70**  | **87.79** | **91.37** |
>
> ### Q3. Regarding comparison between negative reweighting and NIG
>
> **A3.** We sincerely thank the reviewer for this insightful suggestion. As suggested, we conducted controlled ablation experiments on cross-dataset transfer (Table II), with Top-K sampling unchanged throughout.
>
> **Similarity-weighted:** We replace $s_{ij} \cdot u_i$ with $0.5 \cdot s_{ij}$, so the soft label becomes $\lambda_j^{soft} = \lambda_{base} + 0.5 \cdot s_{ij} \cdot w$. The Uncertainty Head and NIG Loss are removed.
>
> **Fixed-threshold:** Samples in Top-K with $s_{ij} > \tau=0.5$ receive $\lambda_j^{soft}= 20\times\lambda_{base}$; others retain $\lambda_{base}$.
>
> Both designs are derived from the analysis in Appendix E, aiming to compress the gradients to ~50% of those of standard negatives.
>
> As shown in Table II, similarity-weighted labeling lacks neighborhood complexity awareness and fails to curb over-repulsion, while fixed-threshold brings only marginal gains. Our NIG method outperforms both by a clear margin, confirming gains stem from uncertainty-aware modeling rather than generic softening. Regarding methods such as Debiased Contrastive Learning assumes randomly distributed false negatives, fundamentally mismatching our structured geographic semi-positives, making direct comparison less meaningful.
>
> **Table II.** Ablation of similarity-weighted and fixed-threshold variants on cross-dataset transfer.
>
> |                     | CVACT->CVUSA |           |           |
> | ------------------- | ------------ | --------- | --------- |
> |                     | R@1          | R@5       | R@10      |
> | Fixed-threshold     | 58.61        | 75.23     | 80.66     |
> | Similarity-weighted | 60.56        | 77.57     | 83.17     |
> | Ours                | **69.34**    | **83.39** | **96.89** |
>
> ### Q4. Regarding theoretical justification and originality
>
> **A4.** We sincerely thank the reviewer for this valuable and constructive feedback. We acknowledge that the set-theoretic framework is explanatory rather than a formal PAC-style bound. As detailed in Appendix E, we provide quantitative gradient-level analysis deriving the explicit gradient reduction of Soft InfoNCE for hard negatives:
>
> $$\Delta g = s_{hard} \cdot 0.5 \cdot w, \quad \rho = 1 - \frac{s_{hard} \cdot 0.5 \cdot w}{p_{hard} - \lambda_{base}}$$
>
> This shows hard negative gradients are compressed to 40%–50%, directly linking our theoretical argument to improved generalization.
>
> Regarding originality, we respectfully note that identifying generalization collapse in CVGL as a previously unrecognized problem, and addressing it via a task-specific design where NIG-derived $u_i$ modulates  soft labels to suppress hard negative gradients, achieving an 18% average R@1 gain on zero-shot transfer, represents a meaningful contribution beyond component assembly.

---

> > ### Author Rebuttal · Reviewer_uyC3 · 2026-04-03
> >
> > The rebuttal has meaningfully alleviated some of my concerns, I update my score.

---

> > > ### Author Response · Authors · 2026-04-04
> > >
> > > We sincerely thank the reviewer for the positive feedback and for updating the score. We are glad that our rebuttal has helped clarify the concerns, and we greatly appreciate your time and consideration.

---

### Official Review · Reviewer_jp6B · 2026-03-11

**Soundness:** 3
**Presentation:** 2
**Significance:** 2
**Originality:** 4
**Overall Recommendation:** 4
**Confidence:** 3

**Summary:**

The paper proposes a model to solve the cross-view geo-localization tasks, i.e., retrieve the satellite image given a street query. The method introduces an uncertainty-aware framework that predicts Normal-Inverse-Gamma parameters for each positive sample using an Uncertainty Head, then uses the resulting uncertainty to soften negative labels in a Soft InfoNCE loss. Experiments show improved retrieval performance, with especially large gains on transfer settings.

**Compliance With Llm Reviewing Policy:**

Affirmed.

**Final Justification:**

After reviewing the paper and the rebuttal, I maintain my recommendation of Weak Accept (4). Overall, the manuscript's major contribution is introducing an uncertainty-aware contrastive learning framework that adaptively modulates the repulsion strength of negative samples via NIG-based modeling.

**Originality, Significance and Clarity**. In terms of originality, the idea of leveraging Deep Evidential Regression to model per-sample uncertainty and guide soft labeling is well-motivated. For significance, the problem is important, and the reported gains, especially on cross-dataset transfer, are substantial and aligned with the paper’s motivation. Regarding clarity, while the initial submission had some ambiguities (e.g., the inputs to the Uncertainty Head), the rebuttal clearly resolved these and improved my understanding.
**Rebuttal**. Importantly, the rebuttal addressed my main concerns. The authors clarified the inputs to the Uncertainty Head and provided additional ablations, further disentangling key components such as similarity-based soft labeling and the NIG mechanism. The comparison against a DINOv3-based baseline is particularly helpful, demonstrating that the improvements are not solely due to the selected backbone. These additions strengthen the causal attribution of the gains and increased my confidence in the method. In summary, the rebuttal reinforced my original assessment and slightly increased my confidence.

**Key Questions For Authors:**

- Please clarify the exact inputs to the Uncertainty Head. Does it attend only over the positive image’s spatial tokens (same view) or also over features of the mined top-K negatives?
- Can you provide the more detailed ablation results by isolating the components mentioned above (Strengths And Weaknesses -  Weakness - 2)?
- Can you provide results for a direct same-backbone baseline, trained with the exact same DINOv3-ViT-B setup but without the uncertainty mechanism, especially for the transfer tasks in Tab.5? If those gains remain comparably large, I would view the paper much more favorably.

**Limitations:**

yes

**Strengths And Weaknesses:**

**Strengths**
- The paper tackles an important problem, cross-view geo-localization (CVGL), which is important for autonomous navigation. The paper targets a real and important pain point in CVGL, i.e., transfer generalization despite high in-domain retrieval scores.
- The proposed pipeline is easy to follow at a high level. Figure 2 is helpful and makes the interaction between Top-K mining, the uncertainty head, NIG fitting, and soft-label generation reasonably clear.
- The empirical results on standard benchmarks are strong. In particular, Table 1 and Table 2 show consistent improvements over prior reported methods, and Table 5 reports very large gains on cross-dataset transfer, which aligns with the paper’s main motivation.

**Weaknesses**
- Ambiguity in the Uncertainty Head inputs. Sec.3.2 Line ~218 says "We use positive sample’s cls token as the query and the spatial tokens as keys and values ...", but Fig.2 Caption Line ~186 and the figure say "The positive sample’s similarities to these Top-K negatives..."?
- Tab.3 only removed four attention metrics and compared two encoders. It remains unclear whether the gains stem from the DER/NIG mechanism, simple soft labeling, Top-K mining, or interactions among these factors. This is crucial because the paper contributes a principle-based uncertainty perception method rather than a simple training adjustment.
- The paper assumes per-sample Top-K similarities are Gaussian in Eq. 6. The Fig.7 is an aggregate histogram which does not validate the conditional per-positive modeling assumption actually used by the method.
- Important details such as runtime, robustness experiments, and some hyperparameter derivations are deferred out of the main text.
- Some VERY minor presentation issues, e.g., incorrect header and wrong font of $L^{soft}_{infoNCE}$ (which should be $\mathcal{L}$) in Fig.2, etc. These issues will not affect my score.

---

> ### Author Rebuttal · Authors · 2026-03-29
>
> ## Response to Reviewer 1 (jp6B)
>
> We sincerely thank **Reviewer jp6B** for their careful reading and insightful comments. The feedback has been invaluable in improving the quality of our manuscript. We have carefully addressed all the concerns raised, and the corresponding revisions have been incorporated into the revised manuscript. The major changes are summarized as follows:
>
> ### Q1. Regarding the inputs to the Uncertainty Head
>
> **A1.** We clarify that the Uncertainty Head takes only the positive sample's spatial tokens and cls token as input. The features of Top-K negatives enter the pipeline solely as scalar similarity observations $t_{ij}$ used to train the NIG distribution, not as attention inputs. The uncertainty $u_i$ is designed to capture the visual environment complexity around the positive sample's geographic location — specifically, the ambiguity of its internal spatial semantic distribution — rather than depending on the specific features of negatives in the current batch. Relying on negative features would introduce batch dependency, reduce the stability of uncertainty estimation, and incur additional computational overhead. By using $t_{ij}$ as observations to train the NIG distribution, the head learns to predict neighborhood complexity directly from the positive sample's internal features.
>
> ### Q2. Regarding additional ablation studies and encoder comparison
>
> **A2.** We thank the reviewer for this constructive suggestion. As suggested, we conducted more detailed ablation and comparison experiments. Specifically, keeping the Top-K mining strategy unchanged, we removed the NIG-trained Uncertainty Head and assigned soft labels based on similarity alone. To ensure a fair comparison, we replaced $s_{ij} \cdot u_i$ with $0.5 \cdot s_{ij}$, where 0.5 is chosen to compress gradients to approximately 50% of their original magnitude (see Appendix E for details). Results are shown in Table I. Assigning soft labels purely through similarity fails to simultaneously mitigate over-repulsion of hard negatives and maintain effective repulsion of ordinary negatives, making model performance sensitive to batch composition. This validates the necessity of our NIG modeling. For similarity weighted, we observe that Top-K has limited impact on performance, as low-similarity negatives yield soft labels close to $\lambda_{base}$; this further highlights the importance of Top-K in fitting the NIG distribution and reducing computational complexity. Furthermore, to verify the contribution of DINOv3, we replaced the baseline backbone with DINOv3-ViT-B (equivalent to our method without the NIG component) and evaluated on cross-dataset transfer, as shown in Table II. The results demonstrate that our method still achieves an average performance improvement of 14% over the DINOv3-based baseline.
>
> **Table I.** Ablation of the Uncertainty Head on cross-dataset transfer (CVACT $\rightarrow$ CVUSA).
>
> | Setting             | R@1   | R@5   | R@10  |
> | ------------------- | ----- | ----- | ----- |
> | Similarity-weighted | 60.56 | 77.57 | 83.17 |
> | Ours                | 69.34 | 83.39 | 96.89 |
>
> **Table II.** Ablation of the backbone (DINOv3-ViT-B) on cross-dataset transfer.
>
> | Method                     | CVACT->CVUSA |       |       | CVUSA->CVACT_val |       |       | CVUSA->CVACT_test |       |       |
> | -------------------------- | :----------- | ----- | ----- | ---------------- | ----- | ----- | ----------------- | ----- | ----- |
> |                            | R@1          | R@5   | R@10  | R@1              | R@5   | R@10  | R@1               | R@5   | R@10  |
> | baseline with dinov3-vit-b | 56.96        | 77.43 | 83.17 | 68.55            | 87.07 | 90.70 | 40.60             | 71.02 | 77.67 |
> | Ours                       | 69.34        | 83.39 | 96.89 | 82.01            | 93.80 | 98.08 | 54.99             | 83.46 | 98.00 |
>
> ### Q3. Regarding the Gaussian assumption
>
> **A3.** We thank the reviewer for this careful observation and agree that Fig. 7 does not directly prove the per-sample conditional Gaussian assumption, and as suggested we will revise the wording to "approximately Gaussian" in the final version. Nevertheless, Fig. 7 provides meaningful indirect support: since our aggregate is a mixture of per-sample distributions, a clean Gaussian-shaped mixture constrains the per-sample components to be approximately homogeneous Gaussian — non-Gaussian components would typically produce multi-modal or heavy-tailed aggregate shapes. The NIG loss further constrains only the first and second moments, conferring robustness to mild per-sample deviations, and consistent gains across benchmarks validate this assumption in practice.
>
> ### Q4. Regarding writing details and important content revisions
>
> **A4.** We sincerely appreciate the reviewer's suggestions. We will carefully review and refine all writing details in the final version, and will move key experimental analyses and results from the appendix into the main text.

---

> > ### Author Rebuttal · Reviewer_jp6B · 2026-04-02
> >
> > Thank the authors for their detailed and professional response. The additional empirical results address my concerns, and the responses clarify the ambiguities. In this case, I will maintain my positive score.

---

> > > ### Author Response · Authors · 2026-04-04
> > >
> > > We sincerely thank the reviewer for the thoughtful re-evaluation and generous recognition of our efforts. We are truly grateful for the time and care invested in reviewing our work, and are glad the additional empirical results and clarifications were helpful. Your constructive feedback has substantially strengthened our manuscript.

---

### Decision · Program_Chairs · 2026-04-30

**Decision:**

Accept (regular)

**Comment:**

This paper   introduces an uncertainty-aware contrastive learning framework for Cross-view geo-localization (CVGL). It targets a real and important pain point in CVGL, i.e., transfer generalization despite high in-domain retrieval scores. Empirical results on standard benchmarks confirm the intuition and show consistent improvements over prior reported methods, and large gains on cross-dataset transfer.

The specific formulation is very similar to a concurrent (slightly earlier work with different image retrieval applications) from NeurIPS 2025 "Towards Robust Uncertainty Calibration for Composed Image Retrieval," including
1) The derivation of total uncertainty  via the Normal-Inverse-Gamma distribution
2) The specific penalty term for erroneous evidence